# Ultrafast tunable lasers using lithium niobate integrated photonics

Viacheslav Snigirev[1,2,6], Annina Riedhauser[3,6], Grigory Lihachev[1,2,6], Mikhail Churaev[1,2,6], Johann Riemensberger[1,2,4], Rui Ning Wang[1,2], Anat Siddharth[1,2], Guanhao Huang[1,2], Charles Möhl[3], Youri Popoff[3,5], Ute Drechsler[3], Daniele Caimi[3], Simon Hönl[3], Junqiu Liu[1,2], Paul Seidler[3✉] & Tobias J. Kippenberg[1,2✉]

Early works[1] and recent advances in thin-film lithium niobate (LiNbO$_3$) on insulator have enabled low-loss photonic integrated circuits[2,3], modulators with improved half-wave voltage[4,5], electro-optic frequency combs[6] and on-chip electro-optic devices, with applications ranging from microwave photonics to microwave-to-optical quantum interfaces[7]. Although recent advances have demonstrated tunable integrated lasers based on LiNbO$_3$ (refs. [8,9]), the full potential of this platform to demonstrate frequency-agile, narrow-linewidth integrated lasers has not been achieved. Here we report such a laser with a fast tuning rate based on a hybrid silicon nitride (Si$_3$N$_4$)–LiNbO$_3$ photonic platform and demonstrate its use for coherent laser ranging. Our platform is based on heterogeneous integration of ultralow-loss Si$_3$N$_4$ photonic integrated circuits with thin-film LiNbO$_3$ through direct bonding at the wafer level, in contrast to previously demonstrated chiplet-level integration[10], featuring low propagation loss of 8.5 decibels per metre, enabling narrow-linewidth lasing (intrinsic linewidth of 3 kilohertz) by self-injection locking to a laser diode. The hybrid mode of the resonator allows electro-optic laser frequency tuning at a speed of $12 \times 10^{15}$ hertz per second with high linearity and low hysteresis while retaining the narrow linewidth. Using a hybrid integrated laser, we perform a proof-of-concept coherent optical ranging (FMCW LiDAR) experiment. Endowing Si$_3$N$_4$ photonic integrated circuits with LiNbO$_3$ creates a platform that combines the individual advantages of thin-film LiNbO$_3$ with those of Si$_3$N$_4$, which show precise lithographic control, mature manufacturing and ultralow loss[11,12].

Lithium niobate (LiNbO$_3$) is an attractive material for electro-optic devices and has been widely used for many decades. It exhibits a wide transparency window from ultraviolet to mid-infrared wavelengths and has a large Pockels coefficient of 32 pm V$^{-1}$, enabling efficient, low-voltage and high-speed modulation. Integrated photonics based on materials exhibiting the Pockels effect—such as aluminium nitride[13]—have been demonstrated before, but only recently for LiNbO$_3$ (ref. [14]). Following the commercial availability of LiNbO$_3$ on insulator via wafer bonding and smart-cut, there has also been substantial progress in the etching of low-loss LiNbO$_3$ waveguides, culminating in ring resonators with an intrinsic $Q$-factor of $10 \times 10^6$ (ref. [2]). The majority of these achievements have utilized argon ion beam etching to manufacture partially etched ridge waveguide structures, which enabled modulators operating at complementary metal–oxide–semiconductor (CMOS) voltages[4], quadrature-phase-shift-keying modulators[15] and electro-optic frequency combs[6]. Furthermore, the platform has provided a route to creating interfaces using cavity electro-optics that efficiently couple microwave-to-optical fields[7]. In addition to direct etching, heterogeneous integration of LiNbO$_3$ chiplets onto silicon nitride (Si$_3$N$_4$)[10] or silicon[16] photonic integrated circuits (PICs) has recently been demonstrated.

Beyond applications for electro-optic modulators, a LiNbO$_3$ integrated photonics platform with a large Pockels coefficient and low propagation loss fulfils all the requirements for realizing integrated narrow-linewidth and frequency-agile laser sources, which feature ultrafast, linear and mode-hop-free tuning. Although integrated lasers have made major advances recently, culminating in hybrid self-injection locked lasers based on high-$Q$ Si$_3$N$_4$ integrated microresonators that reach fibre laser coherence[17,18], that is, subhertz Lorentzian linewidth, these lasers lack fast frequency actuation. Although integrated narrow-linewidth lasers with similar performance have recently been demonstrated using monolithically integrated piezoelectrical stress-optical actuation that is flat and with megahertz actuation bandwidth[19,20], lasers based on LiNbO$_3$ integrated photonic circuits have the potential for vastly faster tuning, with flat frequency response, at substantially lower drive voltages, and do not exhibit excitations of parasitic vibrational modes of the photonic chip, as in the case of piezoelectrical actuation. An electrically pumped hybrid LiNbO$_3$/III–V laser has been demonstrated using a Vernier-filter-based scheme[8,9], but has not yet achieved this capability. Lasers based on LiNbO$_3$ photonic integrated circuits have the potential to realize a host of laser

[1]Institute of Physics, Swiss Federal Institute of Technology Lausanne (EPFL), Lausanne, Switzerland. [2]Center for Quantum Science and Engineering, EPFL, Lausanne, Switzerland. [3]IBM Research - Europe, Zurich, Ruschlikon, Switzerland. [4]Deep Light SA https://deeplight.pro/. [5]Integrated Systems Laboratory, Swiss Federal Institute of Technology Zurich (ETH Zürich), Zurich, Switzerland. [6]These authors contributed equally: Viacheslav Snigirev, Annina Riedhauser, Grigory Lihachev, Mikhail Churaev. ✉e-mail: pfs@zurich.ibm.com; tobias.kippenberg@epfl.ch

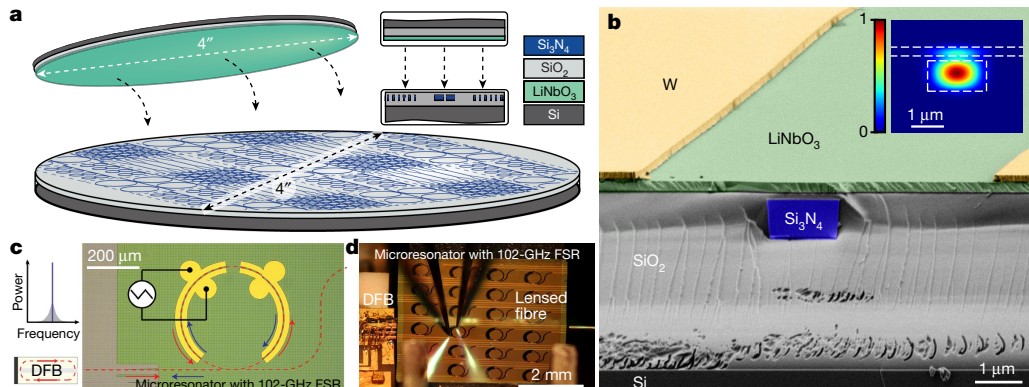

**Fig. 1 | Heterogeneous, low-loss, Si₃N₄–LiNbO₃ photonic integrated platform for fast tunable self-injection-locked lasers. a**, Schematic illustration of the heterogeneous Si₃N₄–LiNbO₃ platform realized by heterogeneous integration of a 4″ (100 mm) thin-film LiNbO₃ wafer onto a 4″ Si₃N₄ wafer, with cross-sections of both wafers. **b**, False-colour SEM image of a heterogeneous Si₃N₄–LiNbO₃ waveguide cross-section. The original SEM image data are shown in Extended Data Fig. 1. Inset: a finite-difference time-domain simulation of the spatial distribution of the hybrid transverse electric mode's electric-field amplitude with 12% participation in LiNbO₃, electric-field maximum is coloured in red and minimum in blue. **c**, Schematic illustration of the self-injection locking principle. The optical path is marked with the dashed red line. The red arrow shows the forward optical wave and the blue arrow shows the reflected optical wave from a microresonator. Laser wavelength tuning is achieved by applying a voltage signal (for example, a linear ramp) on the tungsten electrodes. The structures in yellow are the tungsten electrodes. **d**, Photo of the set-up with a DFB laser butt-coupled to a heterogeneous Si₃N₄–LiNbO₃ chip (sample D67_01b C16 WG 4.2). A pair of probes touch the electrodes for electro-optic modulation, and a lensed fibre collects the output radiation.

structures, such as widely tunable Vernier lasers or mode-hop-free lasers for a multitude of applications, including frequency-modulated continuous-wave (FMCW) light detection and ranging (LiDAR)[21], optical coherence tomography, frequency metrology or trace-gas spectroscopy[22], which utilize both frequency agility and narrow linewidth. Here we demonstrate LiNbO₃-based integrated lasers that achieve narrow linewidth (kilohertz level) while exhibiting extreme frequency agility, allowing a petahertz-per-second tuning rate. This is achieved on a heterogeneously integrated platform combining ultralow-loss Si₃N₄ photonic waveguides[23] with thin-film LiNbO₃ by wafer-scale bonding[24]. Our hybrid platform uses a Si₃N₄–LiNbO₃ chip that is butt-coupled to an indium phosphide (InP) distributed feedback (DFB) diode laser. The Si₃N₄ photonic integrated circuits are manufactured using the photonic Damascene process[23] and feature tight optical confinement, ultralow propagation loss (<2 dB m⁻¹), low thermal absorption heating and high-power handling. They can be manufactured at the wafer scale with high yield and are already available from a commercial foundry. Additional advantages of the Si₃N₄ platform include low gain from the Raman and Brillouin nonlinearities and radiation hardness. This heterogeneous Si₃N₄–LiNbO₃ platform enables high-$Q$ microresonators with a median intrinsic cavity linewidth of 44 MHz, provides a near-unity yield of bonded devices, and exhibits low, compared with LiNbO₃ ridge waveguides, insertion loss of 3.9 dB per facet[24]. In addition, the heterogeneous Si₃N₄–LiNbO₃ platform does not exhibit bend-induced mode mixing owing to the birefringence, as is typically the case for LiNbO₃ ridge waveguides. Combining the unique properties of both materials into a single heterogeneous integrated platform enables laser self-injection locking with two orders of magnitude of laser frequency noise reduction and a petahertz-per-second frequency tuning rate.

## Heterogeneous integration of LiNbO₃ on Si₃N₄ PIC

Our fabrication method combines the processes of photonic Damascene Si₃N₄ waveguide fabrication with wafer-scale bonding[25] to enable electro-optic modulation on passive, ultralow-loss Si₃N₄, as schematically depicted in Fig. 1a. Our process starts with the fabrication of a patterned and planarized Si₃N₄ substrate using the photonic Damascene process (see details in Methods). A silicon dioxide (SiO₂) interlayer is deposited on the substrate, followed by densification. The interlayer is then polished to reduce the remaining topography and set the desired thickness. A root-mean-square roughness of less than 0.4 nm over an area of a few square micrometres and of only a few nanometres over an area of several hundred micrometres are necessary for bonding[24]. Next, a few-nanometre-thick alumina layer is deposited by atomic layer deposition on both the donor (LiNbO₃ on insulator) and the acceptor (planarized Si₃N₄ photonic circuit containing 100-mm wafer) wafers before contact bonding and donor-wafer removal. Tungsten electrodes are then manufactured by sputtering and reactive ion etching. At this point, the areas of the coupling facets and the tapered sections of the Si₃N₄ waveguides are cleared from LiNbO₃ by physical etching with argon ions, so that the laser light can first couple into the chip using inverse tapers[26] before entering the transitioning to the LiNbO₃-covered area. Finally, chip release is performed by chip-facet definition by deep SiO₂ and silicon etching, followed by chip separation through backside silicon lapping. Figure 1b depicts a scanning electron microscope (SEM) cross-section of the heterogeneously integrated LiNbO₃-on-Si₃N₄ waveguide with the following layer thicknesses: bottom silica cladding, 4 µm; Si₃N₄, 950 nm; silica top cladding, 150 nm; LiNbO₃, 300 nm; metal electrodes, 200 nm (the original SEM image is shown in Extended Data Fig. 1). The inset of Fig. 1b shows a simulation of the spatial distribution of the electric-field amplitude in the hybrid mode of our device with 12% participation ratio in LiNbO₃. Statistical analysis of the resonator transmission spectra reveals a 44-MHz median intrinsic cavity linewidth corresponding to a $Q$-factor of 4.8 × 10⁶ and linear propagation loss of 8.5 dB m⁻¹ (Extended Data Fig. 8c).

## Laser self-injection locking

Laser self-injection locking is initiated by butt-coupling of an InP DFB diode laser to the heterogeneous Si₃N₄–LiNbO₃ chip (Fig. 1c,d), and tuning the laser current to match the output frequency to the resonance frequency of the heterogeneous Si₃N₄–LiNbO₃ microresonator. The optical backreflection on surface or volumetric inhomogeneities inside the microresonator provides spectrally narrowband feedback to the laser diode by coupling clockwise and anticlockwise propagating modes. Light in the clockwise mode radiates back to the laser bearing the power fraction given by the reflection coefficient $R$, which depends on the interaction strength of the modes and the resonator coupling efficiency.

The laser diode is forced to oscillate at the frequency of the cavity resonance in the self-injection-locked regime. Assuming that the

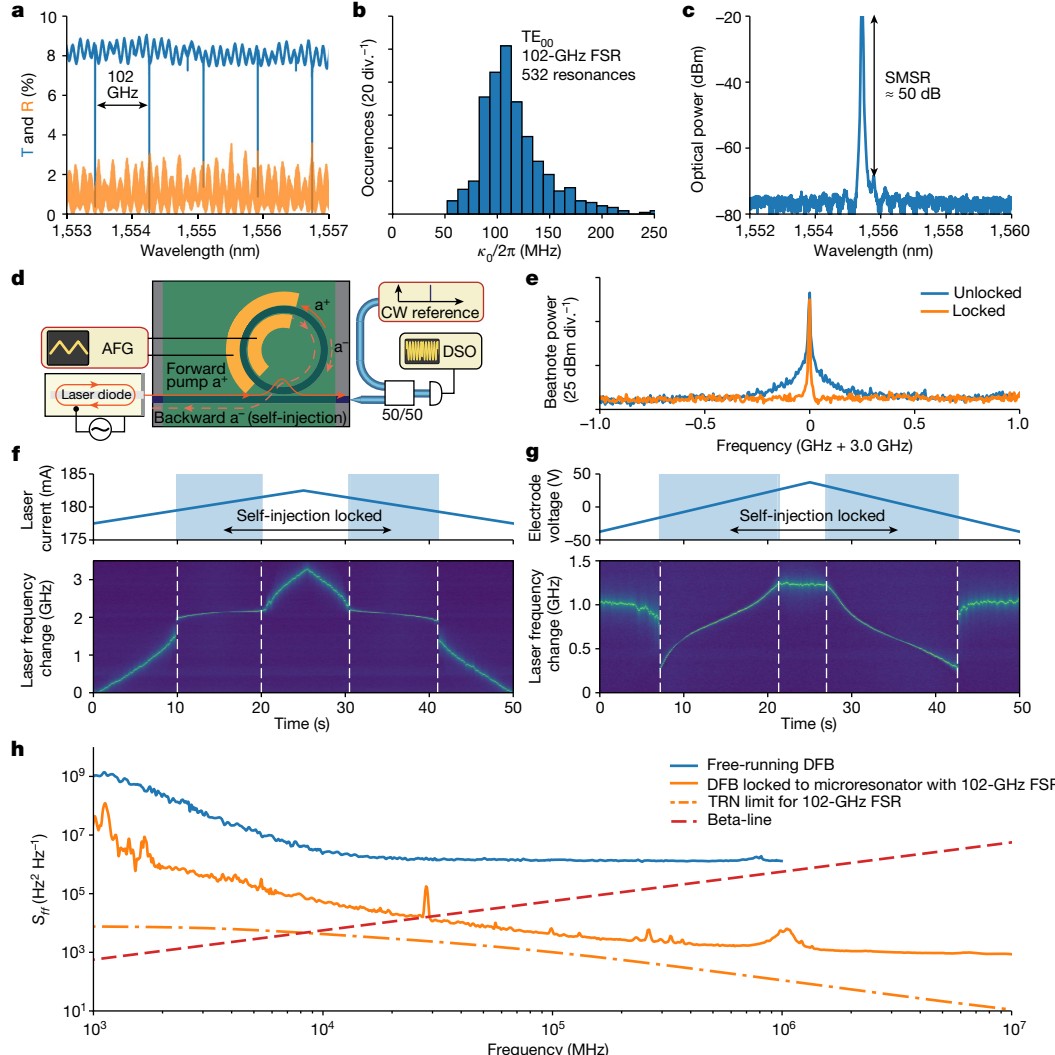

**Fig. 2 | Characterizing the integrated hybrid heterogeneous Si₃N₄–LiNbO₃ platform. a**, Transmission (T, blue) and reflection (R, orange) spectra of a heterogeneous Si₃N₄–LiNbO₃ 102-GHz-FSR microresonator (see Extended Data Fig. 2 for the full dataset). **b**, The histogram shows the distribution of linewidths of 532 resonances for the fundamental transverse electric mode TE₀₀ of the 102-GHz-FSR device with a median linewidth of about 100 MHz, corresponding to a quality factor of $1.9 \times 10^6$ ($\kappa_0$ is the intrinsic cavity decay rate). **c**, Optical spectrum of the free-running DFB laser diode with 50 dB side mode suppression ratio (SMSR). **d**, Experimental set-up for linewidth measurements with the hybrid integrated laser using the heterodyne beatnote method. AFG, arbitrary function generator; DSO, digital storage oscilloscope. The forward pump wave $a^+$ is marked by a solid red line, and the reflected backward wave $a^-$ by a dashed red line. **e**, Comparison of the laser linewidth for the free-running DFB case and

the case where the DFB is self-injection-locked to a heterogeneous Si₃N₄–LiNbO₃ microresonator. **f**, Time–frequency map of the beatnote showing the laser frequency change on linear modulation of the diode current. The white dashed lines mark the boundaries of the self-injection locking bandwidth, where almost no laser frequency change is observed. **g**, Time–frequency map of the beatnote showing the laser frequency change upon linear tuning of the cavity resonance by applying voltage to the electrodes. The DFB current remained fixed in the self-injection locking range. **h**, Frequency noise spectra of the free-running DFB (blue) and the DFB self-injection-locked to the 102-GHz-FSR heterogeneous Si₃N₄–LiNbO₃ microresonator (orange). The evaluated thermo-refractive noise (TRN) limit and the beta-line are given for reference (orange dash-dotted and red dashed lines, respectively).

frequency noise of the laser is white, the frequency noise suppression ratio[27] is:

$$\frac{\delta\omega}{\delta\omega_{\text{free}}} \approx \frac{Q_{\text{DFB}}^2}{Q^2} \frac{1}{16R(1+\alpha_g^2)}, \tag{1}$$

where $\delta\omega_{\text{free}}/2\pi$ is the linewidth of the free-running DFB laser; $\delta\omega/2\pi$ is the linewidth of self-injection-locked DFB laser; $Q_{\text{DFB}}$ and $Q = \omega/\kappa$ are the quality factors of the laser diode cavity and of the microresonator mode, respectively (with $\kappa = \kappa_{\text{ex}} + \kappa_0$, where $\kappa_0$ and $\kappa_{\text{ex}}$ are the intrinsic cavity decay rate and bus–waveguide coupling rate, respectively); and $\alpha_g$ is the phase-amplitude coupling factor. Self-injection locking occurs within a finite frequency interval around the cavity resonance.

The locking bandwidth $\Delta\omega_{\text{lock}}$ is given, assuming large intermodal interaction strength and high coupling efficiency, by[27]:

$$\Delta\omega_{\text{lock}} \approx \sqrt{R(1+\alpha_g^2)} \frac{\omega}{Q_{\text{DFB}}} \tag{2}$$

To strongly reduce the laser linewidth and increase the frequency locking range, a high-$Q$ resonance and strong reflection are desirable. The device used in our experiments features a 102-GHz free spectral range (FSR) and a resonance total linewidth of $\kappa/2\pi = 100$ MHz (Fig. 2a,b) operated close to critical coupling. The intrinsic loss of the microresonator $\kappa_0/2\pi \approx 50$ MHz indicates a waveguide linear propagation loss of 8.5 dB m⁻¹. The power reflection of the device reaches 3%

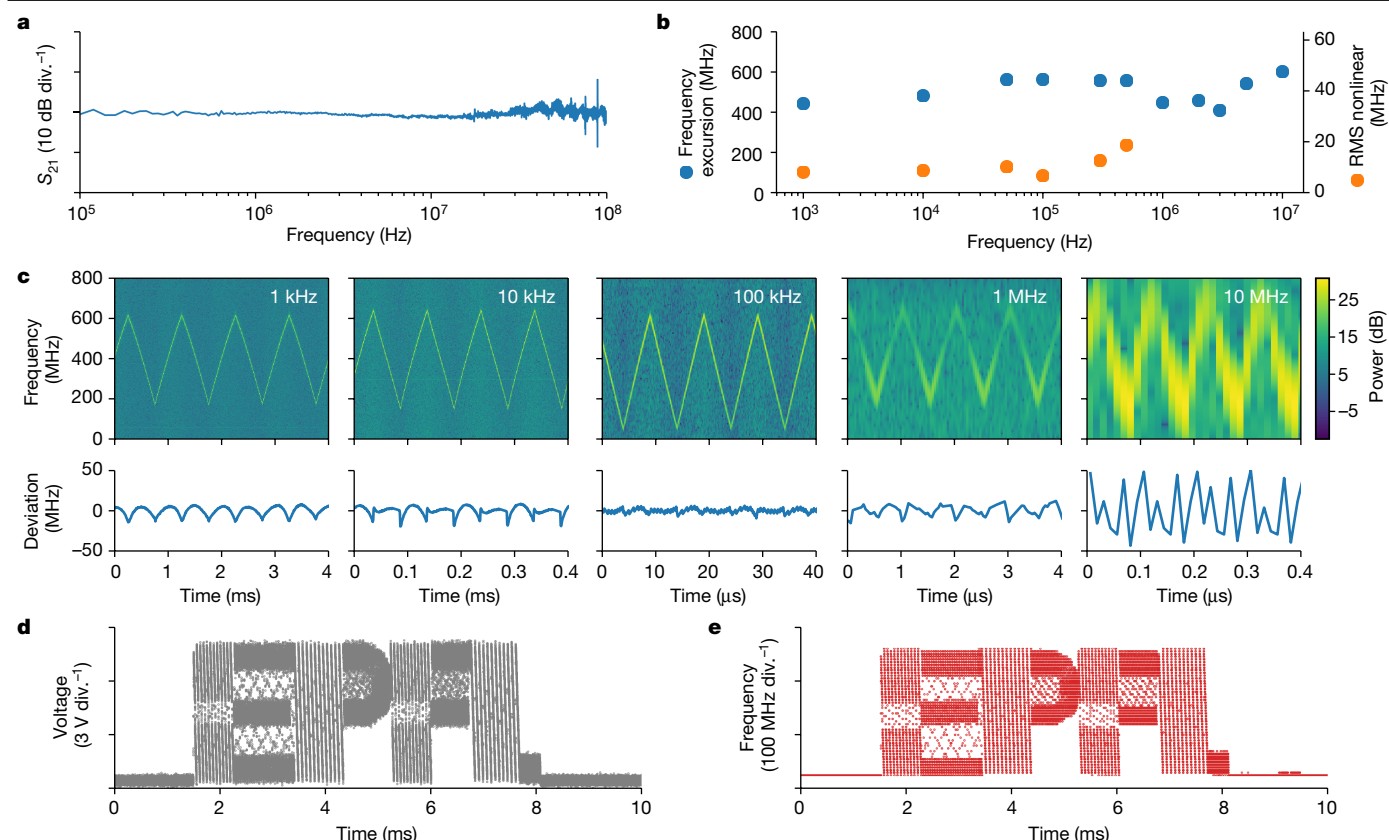

**Fig. 3 | Electro-optic frequency tuning of the self-injection-locked DFB laser. a**, Measured response of the electro-optic modulation for the heterogeneous Si₃N₄–LiNbO₃ device utilizing tungsten electrodes. **b**, Frequency excursion (blue) and the absolute root mean square (RMS) deviation of the measured tuning profile from a perfect triangular ramp (orange). The deviation was calculated as the difference between the experimental data and the least-squares fitting. **c**, Top: time–frequency spectrograms of heterodyne beatnote for modulation frequencies from 1 kHz to 10 MHz. Bottom: the deviation of the experimental tuning data from the least-squares fit for the same modulation frequencies. **d**, Voltage profile applied to the electrodes from an arbitrary waveform generator, resembling the EPFL logo. **e**, Measured laser heterodyne beatnote showing laser frequency evolution in the form of the EPFL logo at a tuning rate of 450 Hz s⁻¹.

(see Fig. 2a and Extended Data Fig. 2 for full spectrum) and features both the narrowband reflection ($R$) of the microresonator and the wideband sinusoidal modulation by spurious reflections from the chip facet, as well as the transitions between the inverse tapers and the heterogeneous Si₃N₄–LiNbO₃ waveguide, which can be mitigated using tapered transitions. Tapered transition in LiNbO₃ also decreases insertion losses to 2.5 dB per facet[24]. The reflection from the chip facets can be reduced by using angled output tapers. Despite the weak back-reflection contrast (see the characterization of other devices from the wafer in Extended Data Fig. 8), injection locking is observed due to the narrow linewidth of the optical resonance. Laser stabilization for any level of intrinsic backscattering can be further improved by introducing an on-chip drop-port-coupled loop mirror. Adjusting optical feedback by tuning the drop-port mirror coupling and feedback phase allows improvement of the locking range and frequency noise suppression[28]. The self-injection-locked DFB emission spectrum (Fig. 2c) indicates a lasing wavelength of 1,555.4 nm with a side-mode-suppression ratio of 50 dB. To characterize laser self-injection locking, a heterodyne beatnote of the unlocked or the locked DFB laser with a reference laser is generated on a fast photodiode and processed using an electrical spectrum analyser (Fig. 2d). We observe beatnote narrowing upon locking of the DFB laser (Fig. 2e). When varying the laser current of the DFB, we found the regions where there is almost no laser frequency tuning because of the self-injection locking (Fig. 2f). To reveal the locking bandwidth, we set the DFB current inside the locked state and scanned the cavity resonance by applying a triangular voltage chirp to the electrodes (Fig. 2g). The self-injection locking is achieved within a frequency span of around 1 GHz; however, linear tuning is observed only within a 600-MHz band due to the low backreflection of the heterogeneous Si₃N₄–LiNbO₃ microresonator.

Next, we measured the (single sided) frequency noise spectral density $S_{ff}(f)$ of the DFB diode laser in the free-running and self-injection-locked regimes (see Methods for details and Fig. 2h for results). The laser self-injection locking suppresses the frequency noise by at least 20 dB across all frequency offsets. We find the intersection point of the frequency noise curve and the beta-line[29] at 30 kHz (Fig. 2h). The full-width at half-maximum (FWHM) linewidth, which is calculated by integration of the frequency noise from the beta-line to the inverse integration time, is 56 kHz at 0.1 ms integration time, 262 kHz at 1 ms and 1.1 MHz at 100 ms. The laser frequency noise reaches a horizontal plateau (white noise floor) of $10^3$ Hz² Hz⁻¹ at a 3-MHz offset, which corresponds to an intrinsic laser linewidth of 3.14 kHz.

## Petahertz-per-second frequency-agile laser tuning

To measure the voltage-to-frequency response of the heterogeneous Si₃N₄–LiNbO₃ microresonator, the signal from a network analyser was applied to the electrodes, and the laser frequency was fixed on the slope of the cavity resonance. This measurement reveals a key advantage of the heterogeneous Si₃N₄–LiNbO₃ platform—the modulation response function for the 102-GHz-FSR microresonator is flat up to the cavity linewidth of 100 MHz (Fig. 3a). To demonstrate the frequency agility of the laser and the response of the laser frequency to a large-amplitude voltage modulation, the DFB laser was self-injection-locked to the

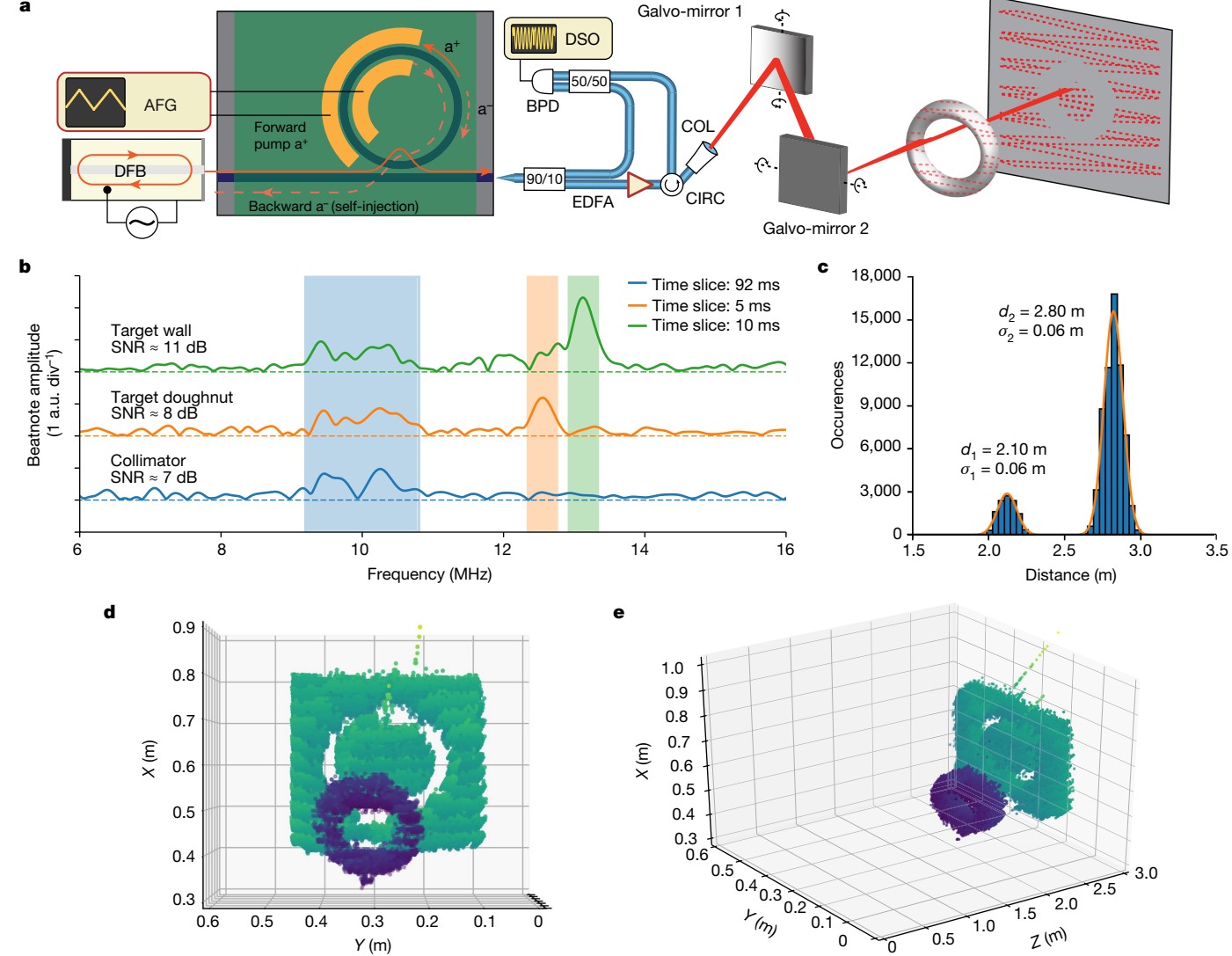

**Fig. 4 | Coherent FMCW LiDAR demonstration using hybrid integrated lasers. a**, Schematics of the experimental set-up for coherent optical ranging based on frequency-modulated continuous wave (FMCW) LiDAR. The output signal of the tunable laser source with a linear frequency chirp is split into two channels for delayed homodyne detection. The signal in the first channel is amplified and, by means of mechanical beam-steering, scans the target. The signal in the second channel is mixed with the fraction of the power of the first channel that was scattered by the target. The beatnote power evolution is recorded by an oscilloscope. AFG, arbitrary function generator; DSO, digital storage oscilloscope; EDFA, erbium-doped fibre amplifier; CIRC, optical circulator; BPD, balanced photodiode; COL, collimator; FPC, fibre polarization controller. **b**, Examples of the delayed homodyne beatnote corresponding to signals from the collimator (blue shaded region in all 3 traces), the doughnut (orange shaded region in orange trace) and the wall (green shaded region in green trace) with the respective SNR values. **c**, Histogram showing the distribution of the calculated values of distance to the target. The two peaks correspond to the reflections from the doughnut and the wall. Both peaks are fitted with a double-Gaussian function with fitting parameters, the mean distance ($d$) and standard deviation ($\sigma$), indicated. **d,e**, Point-cloud representation of the measured target scene from different viewing angles.

microresonator resonance and a triangular voltage signal of 25 Vpp with modulation frequencies ranging from 1 kHz to 10 MHz was applied. Neither signal pre-distortion nor active feedback was applied to the driving signal. The applied voltage modulates the refractive index of $LiNbO_3$ via the Pockels effect and shifts the cavity resonance, forcing the laser to follow the resonance as long as it remains within the overall locking range. To reveal the time-varying frequency-tuning characteristics for large signal modulation in the self-injection-locked state, the heterodyne beatnote of the hybrid integrated laser with the reference laser was recorded on a fast photodiode. Frequency excursions remained on the level of 500 MHz, independently of the modulation frequency, whereas the nonlinearity tended to increase with increased modulation frequency. The minimum nonlinearity of 1% of the frequency excursion is observed at a tuning rate of 100 kHz with a tuning efficiency of 28 MHz V$^{-1}$. The top row in Fig. 3c shows the processed laser frequency spectrograms, which are calculated by time-segmented Fourier transformation, and the bottom row shows the corresponding residuals after a perfect triangular modulation is fit to the data. Figure 3b shows the laser frequency excursion and the root-mean-squared deviation of the measured profiles from a perfect triangular frequency modulation determined by curve fitting. Additional data on tuning efficiency and hysteresis are shown in Extended Data Figs. 3 and 4. The demonstrated frequency excursion of 600 MHz in 50 ns equates to an ultrafast frequency agility of 12 PHz s$^{-1}$.

Although highly linear ramp frequency modulation is essential for FMCW LiDAR application, the frequency can be modulated in an arbitrary manner while preserving a high tuning rate. To illustrate this, we programmed an arbitrary waveform generator to reproduce the logo of

EPFL (Fig. 3d) and applied the signal to the heterogeneous $Si_3N_4$–$LiNbO_3$ device. The laser frequency was again determined by heterodyne beatnote with the reference laser, and the result of the time–frequency analysis is depicted in Fig. 3e, showing a tuning rate of 450 THz s$^{-1}$ and a dwell time between points of 200 ns.

## Optical coherent ranging demonstration

To demonstrate the application potential of our laser, we perform a proof-of-concept optical ranging experiment in a lab environment. The FMCW LiDAR method consists of triangular-shaped frequency modulation of the laser source and delayed homodyne detection with the optical signal reflected from the target. Laser phase noise limits the maximum operating distance and ranging precision in this method. However, a key requirement for FMCW LiDAR at long range is frequency agility, that is, to achieve fast, linear and hysteresis-free tuning[30]. The experimental set-up is depicted in Fig. 4a (see Methods for detailed description). The laser beam is scanned across the target scene by means of two galvo-mirrors with triangular driving signals. We used a polystyrene doughnut-like object and a metal sidewall of a rack enclosure as the target. Both objects were located approximately 3 m away from the collimator. A photograph of the target scene and the beam-scanning pattern are depicted in Extended Data Fig. 5. The beatnote between the signal reflected from the target and the local oscillator is detected with a balanced photodiode and recorded by an oscilloscope. We adjust optical polarization with a fibre polarization controller in the reference arm of the delayed self-homodyne set-up to maximize the signal-to-noise ratio (SNR) of the beatnote signal. Zero-padded short-time Fourier transformation is then applied to the collected oscillogram data to retrieve the beatnote spectrum evolution over 128,000 time slices. The time–frequency spectrograms obtained for both the target and the reference Mach–Zehnder interferometer (MZI) are shown in Extended Data Fig. 6. The MZI was used for distance calibration only (we retrieved a resolution of 15 cm), and neither signal pre-distortion nor active feedback was applied. Figure 4b shows three different time frames with beatnotes of the local oscillator with the reflections from the wall, the doughnut and the collimator, and their respective SNR values. Last, the centre frequencies of the beatnote spectra were identified and mapped into the distance domain using the MZI length as a reference. The resulting distribution of the distance values is plotted as a histogram in Fig. 4c, showing two peaks representing the doughnut at 2.1 m and the wall at 2.8 m. The double-Gaussian fit reveals the statistical distribution of distance values for both objects (Fig. 4c). The point cloud of the three-dimensional optical ranging is inferred from the distance data and the voltage-to-angle conversion of the galvo-mirror controller; it is shown in Fig. 4d,e, where the point colour encodes the distance from the collimator.

In summary, we have demonstrated a heterogeneous wafer-scale platform for electro-optic photonic integrated circuits that integrates ultralow-loss $Si_3N_4$ waveguides and thin-film $LiNbO_3$. We show optical microresonators with 44-MHz median intrinsic cavity linewidth, corresponding to linear propagation losses of 8.5 dB m$^{-1}$, wideband uniform bus–waveguide coupling, and flat electro-optical frequency actuation response up to 100 MHz. Endowing ultralow-loss $Si_3N_4$ photonic integrated circuits with on-chip $LiNbO_3$ electro-optic modulation enables a hybrid self-injection-locked laser with simultaneously narrow linewidth and fast tuning of 12 PHz s$^{-1}$. This laser allows FMCW optical ranging without the need for signal pre-distortion or active feedback and with a resolution of around 15 cm. Detailed comparison with other photonic integrated tunable lasers based on InP chips is given in Extended Data Table 1 and with other integrated $LiNbO_3$ platforms in Extended Data Table 2. With future improvements in photonic circuit design and fabrication, such as reducing the interlayer $SiO_2$ thickness and optimizing electrode positions, we believe that our platform will form the basis of fast tunable lasers with 10-ns-level switching time, mode-hop-free

tuning over tens of gigahertz, and fundamental linewidths below 100 Hz and kilometre-level coherence length. By fully leveraging the high electro-optic coefficient of $LiNbO_3$, with further improvements in photonic integrated circuit design, these devices can operate with complementary metal–oxide–semiconductor-compatible voltages, or achieve millimetre-scale distance resolution. Beyond integrated lasers, the hybrid platform can also be used to realize other functions, such as photonic microwave and millimetre-wave tracking generators[31], switching networks for photonic computing[32], boson sampling[33] and integrated transceivers. Moreover, the wide transparency window of both $LiNbO_3$ and $Si_3N_4$ allows such frequency agility to be extended to other wavelength ranges, such as the mid-infrared or the visible, providing a platform for fast tunable lasers for applications in trace-gas sensing[34].

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

## Methods

### Device fabrication

Our process starts with the fabrication of a patterned and planarized $Si_3N_4$ substrate using the photonic Damascene process[23,35]. Deep-ultraviolet stepper lithography is used to pattern waveguides and microresonators on a silicon substrate with 4-μm-thick thermal wet $SiO_2$. The pattern is then dry-etched into the $SiO_2$ layer to form the waveguide preform, followed by a high-temperature reflow of the waveguide preform[36] to reduce surface roughness. Stoichiometric $Si_3N_4$ is deposited by low-pressure chemical vapour deposition on the patterned substrate, filling the preform and forming the waveguide cores. Chemical mechanical polishing is used to remove excess $Si_3N_4$ and to planarize the wafer top surface. Subsequently, the entire substrate is thermally annealed at 1,200 °C to drive out the residual hydrogen contained in the $Si_3N_4$. The $SiO_2$ interlayer is deposited on the $Si_3N_4$ substrate, densified and subsequently polished using chemical mechanical polishing. Before bonding, a few-nanometre-thick alumina layer is deposited on the donor ($LiNbO_3$ on insulator) and the acceptor ($Si_3N_4$) wafers. After that, both wafers are brought in contact and annealed for several hours at 250 °C. The silicon on the backside of the donor wafer is ground, and the residual silicon after grinding is removed with tetramethylammonium hydroxide wet etching. The thermal $SiO_2$ is wet-etched with buffered hydrofluoric acid. The wafer bonding yield is 100% and we successfully bonded 5 out of 5 wafers during 3 different fabrication runs. A layer of tungsten is sputtered on the $LiNbO_3$ surface and the electrode pattern is transferred in this layer via fluoride-based reactive ion etching. Finally, $LiNbO_3$ is etched to open the chip facet areas to improve the input coupling of the device on the chips by means of argon ion beam etching. The subsequent chip release is performed in three steps: dry etching of chip boundaries in $SiO_2$ with fluorine-based chemistry, further etching of silicon carrier by the Bosch process and backside wafer grinding.

### Laser frequency noise measurements

The free-running DFB performance characterization is shown in Extended Data Fig. 7. We performed heterodyne beatnote spectroscopy[37] beating the reference external-cavity diode laser (Toptica CTL 1550) with the hybrid integrated laser to reveal the frequency noise of the latter. The beatnote of the two signals was detected on a photodiode and its electrical output was then sent to an electrical spectrum analyser (Rohde & Schwarz FSW43). The recorded data for the in-phase and quadrature components of the beatnote were processed by Welch's method[38] to retrieve the single-sided phase noise power spectral density $S_{\phi\phi}$, which was converted to frequency noise $S_{ff}$ using: $S_{ff} = f^2 \times S_{\phi\phi}$. To calculate the laser linewidth, we integrate the frequency noise spectra from the intersection of the power spectral density with the beta-line $S_f(f) = 8\ln2 \times f/\pi^2$ down to the integration time of measurement[29]. The area under the curve $A$ is then recalculated to provide the FWHM measure of the linewidth using: $FWHM = \sqrt{8\ln2 \times A}$. Because a rigorous definition of the optical linewidth does depend on the integration time of the measurement, we evaluate the FWHM linewidth as 56 kHz at 0.1 ms integration time, 262 kHz at 1 ms and 1.1 MHz at 100 ms. The phase noise of the reference laser is determined by another beatnote measurement with a commercial ultrastable laser (Menlo ORS).

### Thermo-refractive noise simulations

The frequency stability of the 102-GHz-FSR heterogeneous $Si_3N_4$–$LiNbO_3$ device is primarily limited by the material refractive index fluctuations due to the relatively large material temperature fluctuations at the microresonator scale, that is, thermo-refractive noise. To quantify the noise level in our system, we follow the approach based on the fluctuation–dissipation theorem (FDT), described in refs. [39–41], that was originally given by Levin and successfully applied to the thermal noise analysis of LIGO's mirrors. As FDT relates fluctuations of a system to how the system dissipates energy, we simulate the noise levels with the finite-element method by testing how the system dissipates in response to a probe force. As the fractional thermo-refractive noise $\frac{\delta\omega}{\omega} = \int d\mathbf{r} q(\mathbf{r})\delta T(\mathbf{r})$ of our device is a weighted average of the temperature fluctuations $\delta T(\mathbf{r})$ determined by the optical field distribution $\mathbf{e}(\mathbf{r})$ with radius vector $\mathbf{r}$, to find out its magnitude at a particular Fourier frequency $f$, we apply a sinusoidal entropy oscillation (energy-conjugated with temperature) at this frequency, with the same weight $q(\mathbf{r})$ mimicking the field distribution, to our system in the simulation. The corresponding power dissipated $W_{diss}$ in the system is retrieved from the simulation and is used to calculate the thermo-refractive noise power spectral density $S_{\frac{\delta\omega}{\omega}}(f)$ at this particular frequency using FDT. The device field distribution and the heat propagation simulated in the described steps are performed on COMSOL Multiphysics.

### Coherent ranging experiment

The laser diode is edge-coupled to the heterogeneous $Si_3N_4$–$LiNbO_3$ chip with 200-nm-thick tungsten electrodes deposited along the $Si_3N_4$ waveguide on $LiNbO_3$. The laser frequency tuning is achieved by locking the laser to a cavity resonance, fixing the DFB current, and tuning the cavity resonance via Pockels effect by a voltage applied to the electrodes. The triangular ramp signal from the arbitrary waveform generator with 0.5 Vpp amplitude and 100-kHz frequency is further amplified up to 25 Vpp by a high-voltage amplifier (Falco Systems) with 5-MHz bandwidth. No additional pre- or post-processing (linearization) was utilized for the laser frequency ramp for the coherent ranging experiment. We used the cavity resonance corresponding to 179-mA DFB current. To calibrate the frequency excursion, the 5% fraction of the optical signal was sent to a reference MZI fibre interferometer. The MZI optical length of 13.18 m was found by an independent measurement involving a tunable diode laser scan calibrated by a frequency comb. Taking the measured MZI optical length and beatnote frequency values, a distance resolution of 15 cm is inferred. Ninety-five per cent of light is split into two paths: the local oscillator path (10%) and the target path (90%). The signal in the target path is amplified by an erbium-doped fibre amplifier (Calmar) from 150 μW up to 4 mW and directed to the collimator with the 8-mm aperture set to match the target distance range of 3 m. We use the galvo scanner (Thorlabs GVS112) for the beam-steering. Two mirrors were controlled by linear ramp signals of 3 Hz and 60 Hz rates with the amplitude and offset values chosen to ensure that the scanning pattern fully covers the target scene. The data for the point cloud were collected within the total time interval of 1.3 s. The frame rate was limited by the galvo scanning speed and the Doppler broadening that is imparted by the rapidly tilting mirrors.

### Scene reconstruction and signal data processing

The data collected in the FMCW LiDAR experiment were subject to digital signal processing steps to locate the scene elements in space. First, the zero-padded short-time Fourier transforms of the beatnote oscillograms from the target and the reference MZI were evaluated. The Blackman–Harris window function was used with the window size set to one period of the frequency-modulated signal. Second, the obtained time–frequency maps were used to search at any given time frame the frequency values corresponding to the beatnote peak. This set was filtered so that only the data points with beatnote amplitudes above some threshold were considered for further analysis. We then subtract the distance from the laser to the collimator so that the point-cloud distance is given with respect to the collimator aperture position. Finally, the frequency data were converted to the distance domain, using MZI length as a reference, and the Cartesian components of each point were computed from the voltage profile applied to the galvo-mirrors.

## Electro-optic $S_{21}$ response measurement

A continuous-wave (CW) laser at 1,550 nm of 300 μW power from an external-cavity diode laser (Toptica CTL 1550) is coupled into the device using a lensed fibre[26]. The input laser is biased at the slope of the optical resonance. A radiofrequency electrical signal of −5 dBm power is applied from port 1 of the network analyser to the electrodes of the device, and the light intensity modulation is detected by a 12-GHz photodiode (New Focus 1544), which is sent back to port 2 of the network analyser.

## Performance comparison

Performance comparison of tunable laser systems[9,17,20,42–54] is presented in Extended Data Table 1. The table compares different tunable laser systems in terms of the frequency tuning range, tuning rate, linearity, optical output power and white frequency noise floor. Performance comparison of different integrated LiNbO$_3$-based platforms[2,55–62] is presented in Extended Data Table 2.

## Data availability

The data used to produce the plots within this paper are available at https://doi.org/10.5281/zenodo.7371066.

## Code availability

The code used to produce the plots within this paper is available at https://doi.org/10.5281/zenodo.7371066.

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

**Acknowledgements** This work was supported by funding from the European Union H2020 Research and Innovation Program under the Marie Sklodowska-Curie grant agreement number 812818 (MICROCOMB) and 722923 (OMT), under the FET-Proactive grant agreement number 732894 (HOT) and grant agreement number 847471 (QUSTEC). It was also supported by funding from the Swiss National Science Foundation under grant agreement number 186364 (QuantEOM) and 201923 (AMBIZIONE), as well as by the Air Force Office of Scientific Research (AFOSR) under award number FA9550-19-1-0250 and by Contract HR0011-20-2-0046 (NOVEL) from the Defence Advanced Research Projects Agency (DARPA). The samples were fabricated in the EPFL centre of MicroNanoTechnology (CMi) and the Binnig and Rohrer Nanotechnology Center (BRNC) at IBM Research. We thank the Cleanroom Operations Team of the BRNC, especially D. Davila Pineda and R. Grundbacher for their help and support.

**Author contributions** M.C. designed the lithography masks and performed PIC simulations. V.S. and G.L. performed experiments with the help of J.R., M.C. and A.S. R.N.W., A.R., C.M. and J.L. developed the processes and fabricated the samples with assistance from S.H. U.D., Y.P., A.R., R.N.W. and J.L. performed the chemical mechanical polishing for bonding. D.C. performed the wafer bonding. V.S., G.L. and J.R. analysed the data. V.S. and G.H. performed thermo-refractive noise limit simulations. V.S., G.L., J.R. and T.J.K. wrote the manuscript with input from A.R., A.S., J.L. and P.S. P.S. and T.J.K. supervised the project.

**Funding** Open access funding provided by EPFL Lausanne.

**Competing interests** T.J.K. is a co-founder and shareholder of LiGenTec SA, a foundry commercializing Si$_3$N$_4$ photonic integrated circuits, as well as DEEPLIGHT SA, a start-up commercializing Si$_3$N$_4$ photonic integrated circuits-based frequency-agile, low-noise lasers.

**Additional information**
**Correspondence and requests for materials** should be addressed to Paul Seidler or Tobias J. Kippenberg.

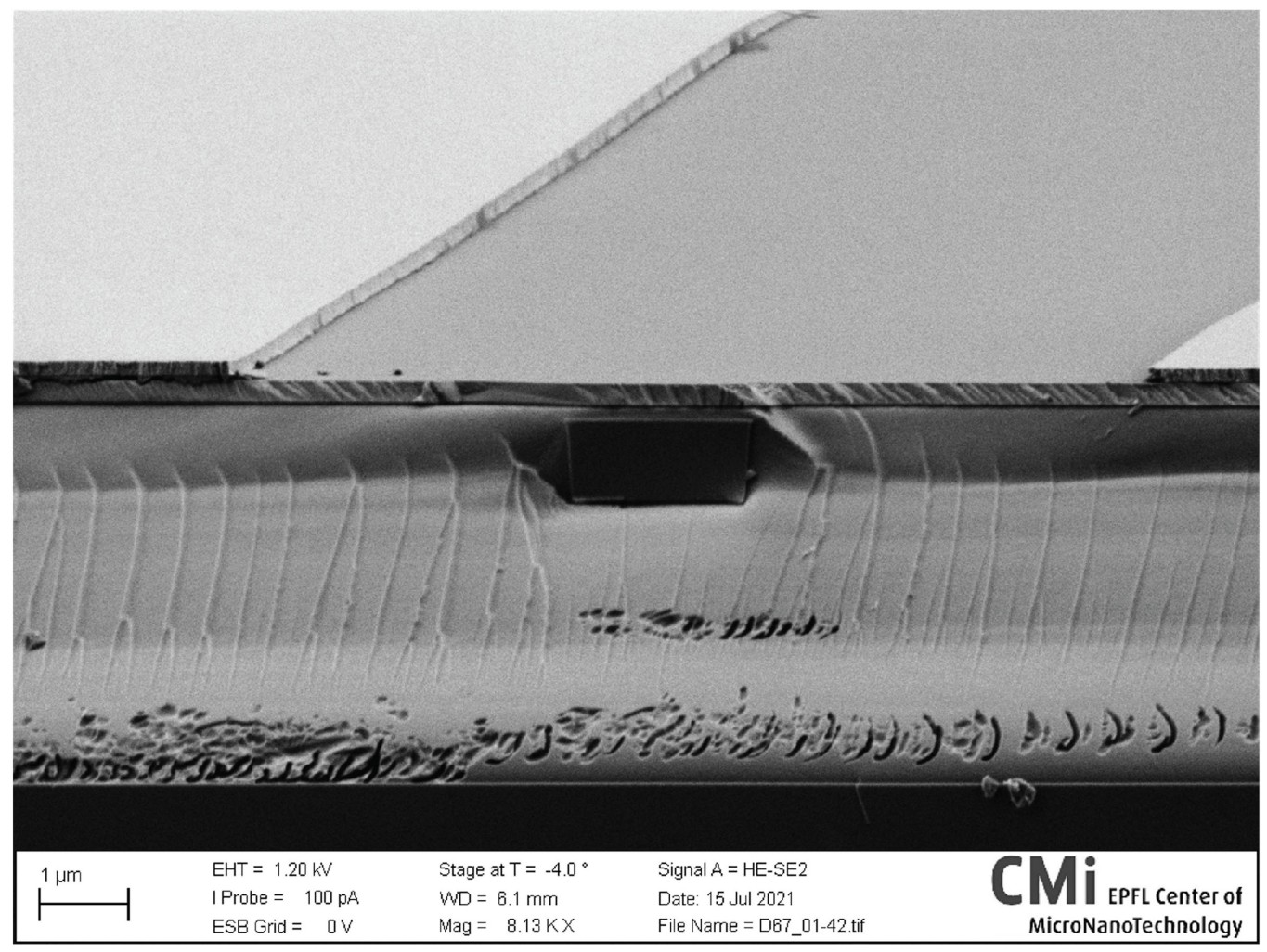

| | | | |
|---|---|---|---|
| 1 μm | EHT = 1.20 kV | Stage at T = -4.0 ° | Signal A = HE-SE2 |
| | I Probe = 100 pA | WD = 6.1 mm | Date: 15 Jul 2021 |
| | ESB Grid = 0 V | Mag = 8.13 K X | File Name = D67_01-42.tif |

**Extended Data Fig. 1 | Scanning electron microscopy image of an heterogeneous $Si_3N_4$-$LiNbO_3$ waveguide.** Original, unprocessed SEM data used to prepare Fig. 1b of the main text.

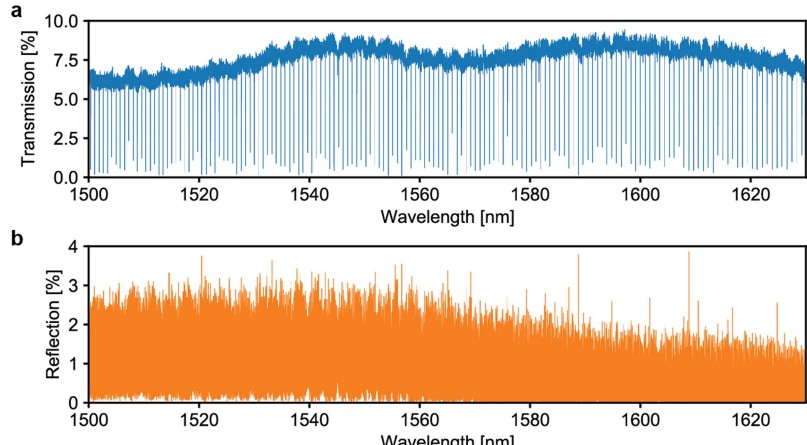

**Extended Data Fig. 2 | Transmission and reflection of a 102-GHz FSR heterogeneous Si₃N₄-LiNbO₃ microring resonator. (a)** Transmission spectrum. **(b)** Reflection spectrum. Data from the sample D67_01b F2 C16 4.3.

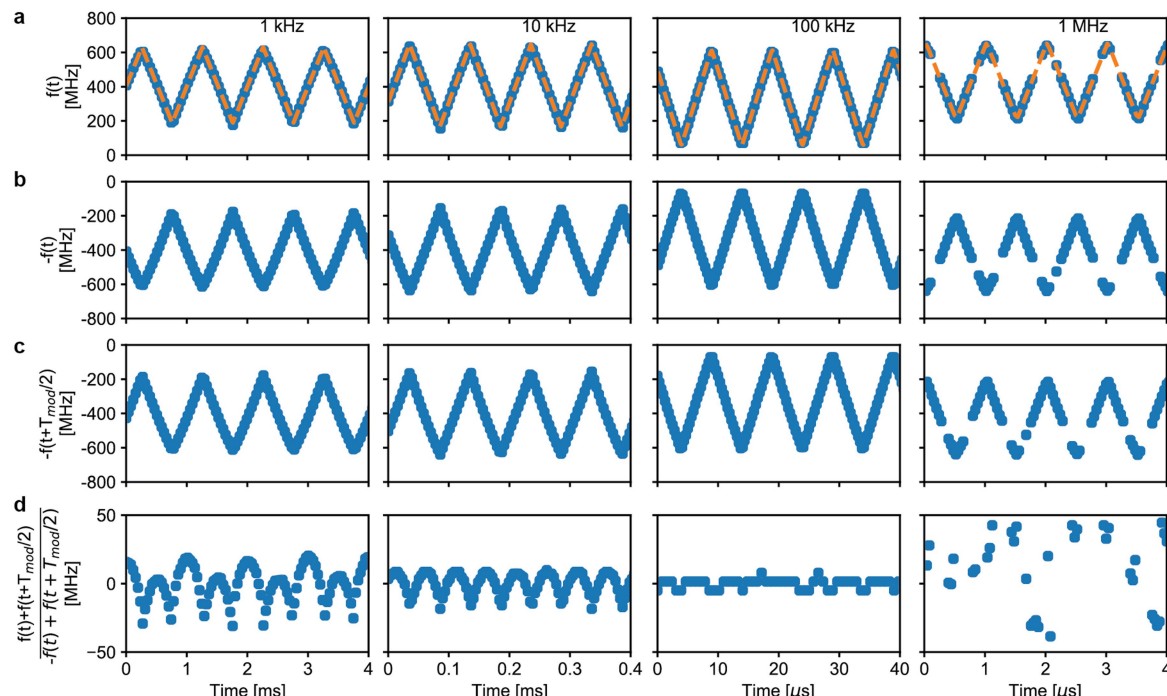

**Extended Data Fig. 3 | Hysteresis evaluation for the DFB laser diode frequency tuning.** The first row **(a)** represents the heterodyne beatnote evolution for the chirping rate of 1 kHz, 10 kHz, 100 kHz and 1 MHz, and its fit with a perfect triangular ramp. The second row **(b)** is the same data but mirrored with respect to a horizontal axis of 0 MHz. In the third row **(c)**, the data is shifted by half a period to the left, such as the up-ramp in the first row becomes the down-ramp in the third and vice versa. In the last row **(d)**, adding the data patterns of the first and the third row and subtracting the mean value from the sum, one observes the hysteresis-induced deviations between the up-ramp and down-ramp.

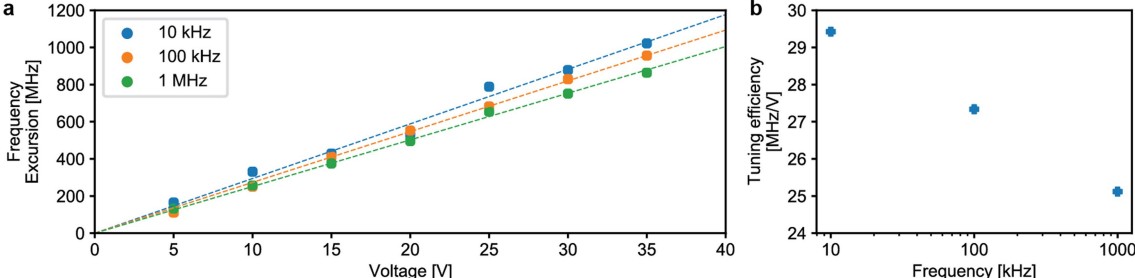

**Extended Data Fig. 4 | Tuning efficiency of heterogeneous Si₃N₄-LiNbO₃ samples. (a)** Applying a triangular ramp voltage waveform to heterogeneous Si₃N₄-LiNbO₃ device's electrodes with selected values of the modulation frequency (10 kHz, 100 kHz and 1 MHz) and gradually increasing the signal peak-to-peak amplitude, one can observe linear growth of the DFB frequency excursion. **(b)** To retrieve the tuning efficiency values, the linear model fit of the data can be performed in the voltages range where the excursion is smaller than the locking bandwidth limit of ~1 GHz (see inset).

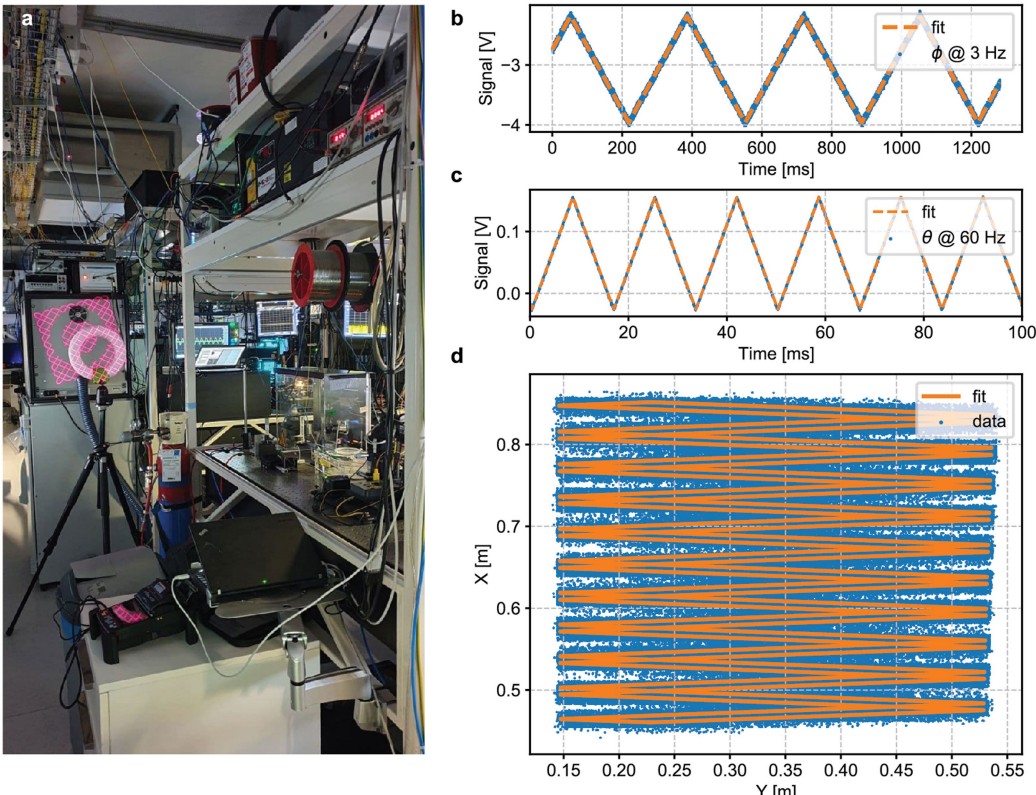

**Extended Data Fig. 5 | FMCW LiDAR set-up. (a)** The photo of the set-up for coherent optical ranging experiment containing the target - a polystyrene doughnut mounted on the stage and an instrument box wall behind, and the scanning pattern of the galvo-mirrors. **(b,c)** The voltage signal profiles applied to two galvo-mirrors enabling two angular degrees of freedom - $\phi$ and $\theta$ - for scanning, and their fits with a perfect triangular ramp. **(d)** The actual data of the scanning pattern and its reconstruction after fitting the angular coordinates.

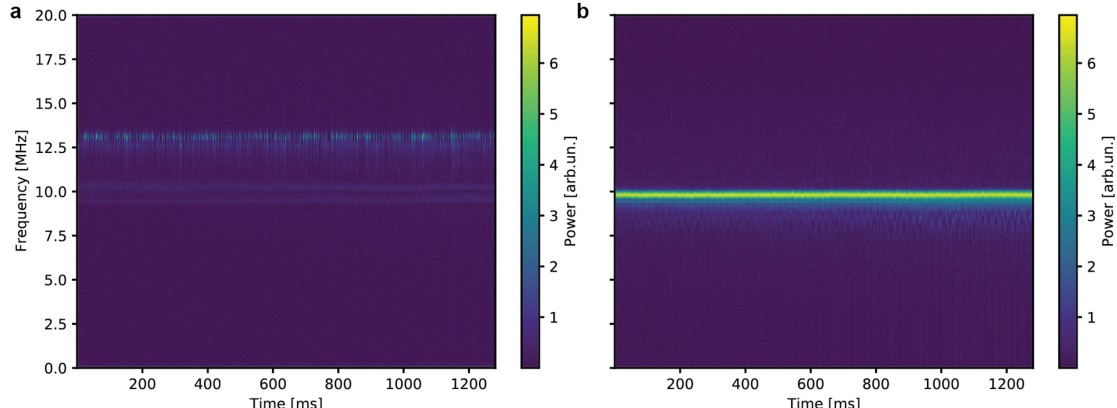

**Extended Data Fig. 6 | Short-time Fourier transform of the delayed homodyne beatnote oscillogram. (a)** Time–frequency map calculated for the target response. **(b)** Time–frequency map for the reference Mach–Zehnder interferometer.

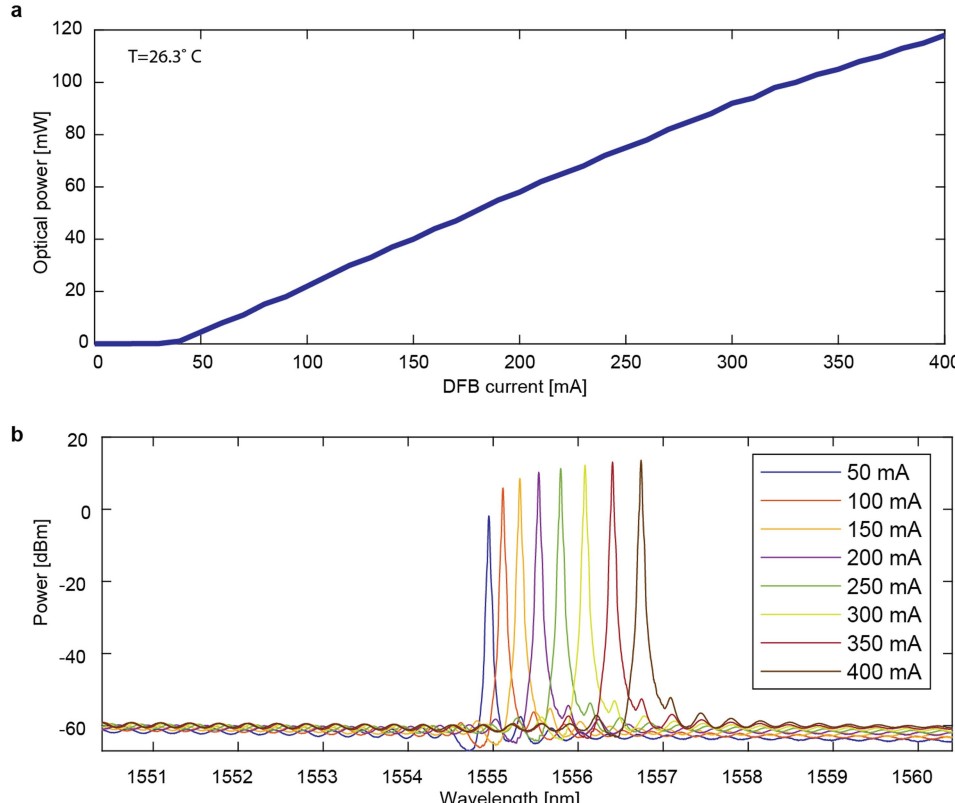

**Extended Data Fig. 7 | Free-running DFB characterization. (a)** Laser diode optical power in free-space vs. diode current. **(b)** Free-running DFB optical spectra at different driving currents.

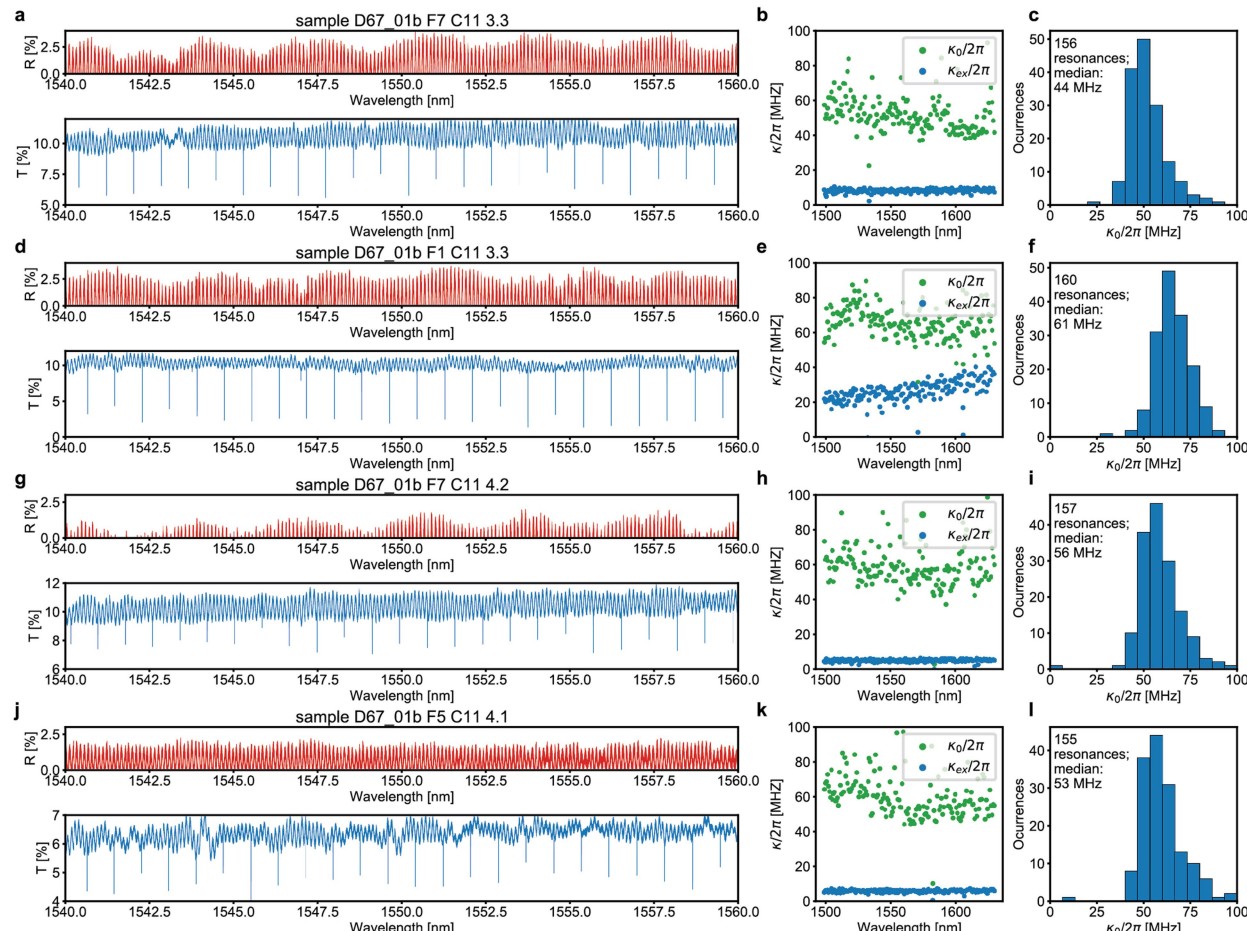

**Extended Data Fig. 8 | Heterogeneous Si₃N₄-LiNbO₃ wafer characterization.** (a,d,g,j) Transmission of bus–waveguide coupled to heterogeneous $Si_3N_4$-$LiNbO_3$ microresonator with free spectral range (FSR) 102 GHz (C11) from 3 fields of the wafer and different microresonators on a chip (F7 WG3.3, F1 WG3.3, F7 WG4.2, F5 WG4.1); (b,e,h,k) Frequency-dependent intrinsic microresonator loss $\kappa_0/2\pi$ (green) and bus–waveguide coupling $\kappa_{ex}/2\pi$ (blue). (c,f,i,l) Histogram of intrinsic microresonator loss rate of wafer D67_01b.

**Extended Data Table 1 | Performance comparison of tunable laser systems**

| Ref. | Active (Gain) element | Passive element | White noise floor | Modulation frequency ($f_{mod}$) | Tuning range (B) | Tuning rate ($f_{mod} \cdot B$) [Hz/s] | Linearity | Optical output power (mW) | FMCW LiDAR demo |
|---|---|---|---|---|---|---|---|---|---|
| 42 | InP RSOA | Si$_3$N$_4$ chip | 40 Hz | n/a (heaters) | 8.7 THz | n/a | n/a | 23 | No |
| 43 | InP die | SOI chip | 220 Hz | n/a (heaters) | 30 GHz (continuous) | n/a | Low | 3.5 | No |
| 44 | GaAs die | SOI chip | 5.3 kHz | n/a (heaters) | 6.02 THz (total) | n/a | Low | 1.5 | No |
| 45 | GaAs die | SOI chip | n/a | n/a (heaters) | 3.1 THz | n/a | Low | 2.7 | No |
| 46 | DFB | SOI chip | n/a | 100 Hz | 64 GHz | $6.4 \times 10^{12}$ Hz/s | High | 1 | Yes |
| 47 | DFB SCL | n/a | 1 MHz | 1 kHz | 100 GHz | $10^{14}$ Hz/s | High (opto-electronic feedback loop) | 40 | Yes |
| 48 | VCSEL | n/a | n/a | 5 kHz | 155 GHz | $7.75 \times 10^{14}$ Hz/s | High (ILC pre-distortion) | n/a | Yes |
| 49 | VCSEL | n/a | 1 MHz | 10 kHz | 11 THz | $1.1 \times 10^{17}$ Hz/s | High (k-point sampling) | 1 | Yes |
| 50 | 12 stitched DFBs | n/a | 3 MHz | 330 Hz | 5.56 THz | $1.8 \times 10^{15}$ Hz/s | High (after linearization) | 12.7 | Yes |
| 51 | InP RSOA | LNOI chip | n/a | n/a (heaters) | 6.4 THz | n/a | n/a | 2.5 | No |
| 9 | InP die | LNOI chip | < 1.5 MHz | n/a | 2.8 THz | n/a | n/a | 0.77 | No |
| 52 | InP DFB | LNOI chip | < 1 MHz | n/a | n/a | n/a | n/a | 19.8 | No |
| 53 | ECDL | LN WGM resonator | 1 MHz | n/a | 28 GHz | n/a | n/a | n/a | No |
| 54 | DFB, Ti:Sa, E-DBR | LN WGM resonator | n/a | 40 MHz | 5 GHz | $40 \times 10^{15}$ Hz/s | n/a | n/a | No |
| 20 | DFB | Si$_3$N$_4$ chip with AIN/PZT | 25 Hz | 800 kHz | 2.1 GHz | $1.7 \times 10^{15}$ Hz/s | High | 1.5 | Yes |
| 17 | DFB | Low confinement Si$_3$N$_4$ | 1.2 Hz | n/a | n/a | n/a | n/a | no data | No |
| This work | InP DFB | LNOD chip | 3 kHz | 10 MHz | 1.2 GHz | $12 \times 10^{15}$ Hz/s | High | 0.15 | Yes |

Frequency tuning range, tuning rate, linearity, optical output power and frequency white noise floor are presented.

**Extended Data Table 2 | Performance comparison of integrated lithium-niobate-based platforms**

| Ref. | Intrinsic Q-factors | Linear optical loss | Statistical analysis | $V_{\pi}L$ product, EO efficiency | PDK | Wafer-level fabrication |
|---|---|---|---|---|---|---|
| This work | $4 \times 10^6$ | 8 dB/m | yes | 30 MHz $\cdot V^{-1}$ ($\approx$ 30 V·cm) | yes | yes |
| 63 | $\approx 10^7$ | 2.7 dB/m | no | not shown | no | yes |
| 55 | no resonators | no data | no | 6.7 V·cm | no | no |
| 56 | $2.5 \times 10^6$ | no data | no | 500 MHz·$V^{-1}$ | no | yes |
| 10 | no resonators | (20± 40 dB/m) | no | not shown | no | no |
| 58 | $1.8 \times 10^6$ | 27 dB/m | yes | not shown | yes | yes |
| 59 | $7.68 \times 10^6$ | 20 dB/m | no | 5.1 V ·cm | no | yes |
| 60 | no resonators | no data | no | 3.1 V·cm | no | yes |
| 61 | no resonators | 70 dB/m | no | 3 V·cm | no | yes |
| 62 | $8.19 \times 10^5$ | 44 dB/m | yes | not shown | no | no |

Intrinsic quality factor, linear optical loss, presence of statistical analysis, $V_{\pi}L$ product, PDK availability, wafer-level fabrication are presented.