## [Peer Review File · Nature]

Manuscript Title: Ultrafast tunable lasers using lithium niobate integrated photonics

Reviewer Comments & Author Rebuttals

Reviewer Reports on the Initial Version:

Referees' comments:

Referee #1 (Remarks to the Author):

Review of Nature 2021-10-16184

This paper reports on the hybrid integration of an injection-locked tunable laser using heterogenous integration of LiNbO₃ with silicon nitride waveguides for the tunable laser cavity and hybrid integration of a butt coupled InP gain block. The laser is then used to demonstrate fast wavelength tuning while maintain certain optical properties and application to laser ranging.

While the work is interesting and of good technical quality, it is not appropriate for publication in Nature. The work does not represent a novel, significant breakthrough, does not report on work of outstanding scientific importance. The submitted work does not reach a conclusion of interest to the Nature interdisciplinary readership. There are major issues with the novelty and claims as well as the presentation of the results. This paper is more appropriate for another optical journal.

1. In the abstract and elsewhere, the use of "hybrid" and "heterogenous" are not clear which can confuse readers who are not central to this area of research. The platform is a hybrid integrated laser, and as such is a very common type with butt-coupled III-IV gain element. While the tuning mechanism is heterogeneously integrated with the waveguide (lithium niobate on silicon nitride), the gain is hybrid integrated (butt coupled). There are many papers with heterogeneously integrated tuning elements. As such, the claim of a heterogeneously integrated lithium niobate laser and hybrid integration will be confusing to the reader and does not represent the level of novelty appropriate for a Nature publication.

a. "Laser self-injection locking is initiated by butt-coupling of an InP distributed-feedback (DFB) diode laser to the LNOD chip (Fig. 1 (c,d))"

b. "This is achieved through heterogeneous integration of ultralow-loss silicon nitride (Si₃N₄) photonic circuits with thin-film LiNbO₃ via direct wafer bonding.

c. The hybrid platform features low propagation loss of 8.5 dB/m, enabling narrow-linewidth lasing (intrinsic linewidth of 3 kHz) by self-injection locking to a III-V semiconductor laser diode."

2. The claim of a first lithium niobate hybrid tunable laser is not accurate. There has been prior publication of such results:

a. Sun Yat-sen University in China and University of British Columbia in Canada claim the first electrically pumped hybrid lithium niobate/III-V laser [Ya Han et al, Optics Letters, v46, p5413, 2021].

3. The reported advancement is not significant enough for a Nature publication. This is acknowledged in the paper. CMOS compatible voltages would be one of the significant results needed to move this field forward.

a. with further improvements in photonic integrated circuit design, these devices can operate with CMOS-compatible voltages, or achieve millimetre-scale distance resolution.

4. While the claim of heterogenous integration of an electrooptically tunable material (lithium niobate) to a waveguide platform (silicon nitride) is accurate, similar results have been published before, by multiple groups including the authors:

a. Low-loss Thin Film Lithium Niobate Bonded on Silicon Nitride Waveguides, Siddhartha Ghosh, Siva Yegnanarayanan, Matthew Ricci, Dave Kharas, and Paul Juodawlkis, CLEO 2020.

b. Heterogeneous integration of lithium niobate and silicon nitride waveguides for wafer-scale photonic integrated circuits on silicon, Optics Letters, Feb. 2017.

5. The waveguide loss and laser performance do not represent a significant breakthrough. There have been multiple publications of injection-locked hybrid integrated lasers, and even of heterogeneous integrated injection-locked lasers, as well as fast tuning in a heterogenous platform:

a. Hertz-linewidth semiconductor lasers using CMOS-ready ultra-high-Q microresonators, Warren Jin, Qi-Fan Yang, Lin Chang, Boqiang Shen, Heming Wang, Mark A. Leal, Lue Wu, Maodong Gao, Avi Feshali, Mario Paniccia, Kerry J. Vahala & John E. Bowers, Nature Photonics volume 15, pages 346–353 (2021)

b. Fully Integrated Photonic Millimeter-Wave Tracking Generators on the Heterogeneous III–V/Si Platform Rui-Lin Chao, Linjun Liang, Jin-Wei Shi, Tin Komljenovic, Jared Hulme, M. J. Kennedy, and J. E. Bowers

6. The performance in terms of optical waveguide loss, resonator Q, and intrinsic linewidth do not represent record values in integrated photonics.

a. “The hybrid platform features low propagation loss of 8.5 dB/m, enabling narrow-linewidth lasing (intrinsic linewidth of 3 kHz) by self-injection locking to a III-V semiconductor laser diode.”

Referee #2 (Remarks to the Author):

A. Summary of the key results

The work submitted for review presents the performance of a hybrid integrated laser source realized combining a Distributed Feedback Grating (DFB) laser in InP with a ring resonator defined in the Lithium Niobate on Damascene Si₃N₄ (LNOD) platform. The external ring resonator cavity can potentially have extremely high quality factor and provides a weak back-reflected wave to the DFB, causing injection locking regime in specific conditions. The laser is tuned in locking regime by changing the resonance frequency of the ring resonator by the electro-optic Pockel effect. The authors claim to reach an ultra-fast tuning speed of 12 PHz/s over a range of 600 MHz, while keeping a laser linewidth of 100 MHz. The laser swept source is then deployed on a coherent LIDAR experiment using a Frequency-Modulated Continuous Wave (FMCW) scheme, demonstrating a resolution of 15 cm in a 3 m range.

B. Originality and significance: if not novel, please include reference

The main novelty, according to the authors, lies in the realization of integrated laser using their LNOD platform. Their platform allows to combine high-confinement silicon-nitride waveguides with the electro-optic material lithium niobate, so to obtain compact, high-quality factor and ultrafast tuning. While the concept is interesting, it is very clear that there is large room of improvement: currently, the performance of such laser does not reach what has been already reported in literature (see Jin, W., Yang, QF., Chang, L. et al. Hertz-linewidth semiconductor lasers using CMOS-ready ultra-high-Q microresonators. *Nat. Photonics* 15, 346–353 (2021). <https://doi.org/10.1038/s41566-021-00761-7>). The main result, being the extremely fast tuning speed, is not enough when the laser linewidth is 100 MHz and the tuning range is limited to 600 MHz, which provides poor resolution in a FMCW LIDAR scheme. Moreover, the deviation is far too high (reaching 100 MHz) to be acceptable in such a narrow range.

C. Data & methodology: validity of approach, quality of data, quality of presentation

The work is well presented, and data are very well represented. All results are exhaustively reported, and the readers have all the elements to draw their conclusions. The paper follows a clear a logical structure. However, there is little comparison with state-of-the-art works. There are notable errors, for example in line 103, there is a contradiction between the Fig. 2d for the direction of back-reflected radiation (counter-clockwise against clockwise). Also, some information is missing: the description of the LNOD platform does not include all the thicknesses of the various layers; the properties of the DFB laser source seed are not clearly stated in the text (RIN, LI curve, grating strength and so on) and they are only reported for a single operating point in Fig. 2c. Also, there is no indication on the output power of the hybrid laser.

D. Appropriate use of statistics and treatment of uncertainties

The experimental data are represented together with their deviation from the intended value. Such data give clear indication of the performance of the device and the range of applicability. However, since the device relies heavily on a structure realised using a custom fabrication process, one should expect a better treatment of the statistics on the ring resonator properties. In particular, the quality factor and the back-reflection coefficient should be reported for a significant number of ring replicas. It is apparent that both the quality factor and the reflection coefficient should be much higher for better performance. Moreover, it is expected that, with the improvement of the fabrication process, the quality factor will be enhanced whereas the back-reflection coefficient, which is triggered by scattering processes introduced by imperfections, will be minimised. What is

the trade-off? Furthermore, there is no mention to the reflection introduced by the butt-coupling and the end-facet on the other side, which inevitably play a role, unless some actions were taken. Finally, the most important result of this work, being the tuning speed at 12 PHz/s, is measured over a range affected by a deviation that could be unacceptable in most applications.

E. Conclusions: robustness, validity, reliability

The performance and operating range of the hybrid laser heavily relies on the fabrication process, which at the current stage, has not developed the required maturity yet to stand-out and make a breakthrough in the realization of optical sources with superior performance. It is not clear what is the level of control on the properties of the fabricated ring resonator, and the whole device.

Furthermore, the performance depends also on the properties of the DFB laser source, for which is unknown the reproducibility. It is also not clear what are the advantages for this device offered by the Damascene process for the definition of the silicon nitride waveguides, which gives the name to the platform.

F. Suggested improvements: experiments, data for possible revision

At the current stage, it would be important to have statistical data on the ring resonator, and different devices realized with different rings. It would be also useful to have the step response of the device to understand the dynamic of the tuning.

G. References: appropriate credit to previous work?

The work provides appropriate credit to previous work for all the concepts described throughout.

H. Clarity and context: lucidity of abstract/summary, appropriateness of abstract, introduction and conclusions

The results are reported in various points of the manuscript, abstract, introduction and conclusions. The abstract however already includes possible future works and improvements that should be exclusive of the conclusions only. In the introduction, final results are included and they can confuse the reader.

Referee #3 (Remarks to the Author):

The authors report on self-injection locking of a DFB laser by an electro-optically tunable micro-resonator based on a lithium-niobate-on-insulator ring cavity. Tuning for about 600 MHz is demonstrated with rates of up to 10 MHz. The system is tried and tested in a frequency-modulated continuous-wave radar system, providing 16 cm depth resolution for a target in 3 m distance.

There is a wealth of applications for tunable lasers with electrical wavelength control. I am confident that laser systems like those demonstrated here will become relevant. The implementation shown here is new, and the experimental work appears to be solid and profound. Thus I am in favor to accept this manuscript for publication in Nature Photonics.

There are, however, some deficiencies that should be resolved before publishing this work. Below I will list my points in the order of occurrence, but let me start with some general statements:

Some relevant citations are missing, being essential to put the work into the right context and to give appropriate credit to previous work.

Some parts of the descriptions are incomplete.

Although I share the enthusiasm for this innovation, one should be realistic which applications such lasers can address and which not. Hence some exaggerations need to be removed and a more realistic discussion about the pending issues would be appreciated.

Line 35: Please do not mention OCT. To my mind it is not feasible that this light source will be applied there. For OCT tuning by hundreds or thousands of GHz is needed, being out of range for the method introduced here. Furthermore, there are established solutions for OCT; there is not much pressure for better lasers in this area.

Line 40: Here it is stated that the first integrated photonics based on lithium niobate has been realized just recently. This is not true and should be corrected. Please see M. A. Aboon and M. M. Fejer, *Opt. Lett.* 22, 151 (1997), G. Schreiber et al., *Proc. SPIE* 4277, 144 (2001) for an early review and Guarino et al., *Nat. Photon.* 1, 407 (2007), just as some examples. I would also call the standard telecom modulators, that are based on waveguides in LiNbO₃, already an integrated photonic solution. Thus there exists already a mass application.

Line 44: Optional, with regard to integrated high-Q LNOI resonators – Referencing Wolf et al., *Optics Express* 25, 29927 (2017)

Line 59: Optional, with regard to piezoelectric tuning – Referencing Ch. Werner et al., *Optica* 4, 1205-1208 (2017) and compare the specs to those of the work under consideration.

Line 59: The closest technical competition to the system presented in this work is the adiabatic electro-optic tuning of laser light with LiNbO₃ micro-resonators, please see Minet et al., *Opt. Express* 28, 2939 (2020). The tuning rate achieved there is 250 PHz/s, instead of 12 PHz/s, as presented in

the work under consideration.

Line 89: What is the purpose of the Al layer? If needed for a stable process: Is there an impact on the optical transparency stemming from an Al-mediated electrical conductivity?

Line 90: The process called “donor-wafer removal” should be explained more thoroughly, maybe in two sentences. What is the yield of the (bonding) process?

Line 119: Since there is no way to vary the coupling strength for a device given, where do you know from that you are close to critical coupling?

Line 147/148: The Toptica laser is sufficiently stable to catch via the heterodyne measurements the stability of the realized, integrated laser? Or has the Menlo laser, mentioned in line 248, been used as a reference?

Line 178: Please write which optical polarization the different beams have and what happens with the polarization in the circuit and which impact this has onto the beat signal. Mixed polarizations will fade the signal of coherent detection.

Line 273: The optical power after (!) the EDFA is just 4 mW. How much is it before? Please add this information to the text. Please consider in the discussion: Optical amplifiers are available just for some few spectral domains. In case that the light source without amplification delivers below 1 mW, most of the applications mentioned in concluding part of the article cannot be realized. Hence the question arises: How can the power be enhanced, without using subsequent amplification? What are the power limits?

Fig. 2e: Where does the “+ 3 GHz” come from?

Fig. 2g: The laser frequency follows the applied field only approximately in a linear manner und this also just in a limited range. Any idea where this comes from? Linearity is crucial for most applications, FMCW LiDAR in particular.

Fig. 3c: From this panel I get the impression, that the best tuning is achieved at 100 kHz, which is surprising. I would suppose that slower is better. Is there an explanation for this?

Fig. 4b: Which times are specified in the inbox?

Karsten Buse

Author Rebuttals to Initial Comments:

A detailed response to all the reviews of the manuscript

We are grateful that three reviewers have seen our manuscript. After reading the comments made by the referees thoroughly, we would like to thank them for the detailed review and suggestions to improve the manuscript. We appreciate that the reviewers gave positive evaluations such as:

“While the work is interesting and of good technical quality ...” (Referee #1)

“The work is well presented, and data are very well represented. All results are exhaustively reported, and the readers have all the elements to draw their conclusions” (Referee #2)

“There is a wealth of applications for tunable lasers with electrical wavelength control. I am confident that laser systems like those demonstrated here will become relevant. The implementation shown here is new, and the experimental work appears to be solid and profound.” (Referee #3)

Before responding individually to each Reviewer, we would like to first make an Executive Response to all reviewers to highlight the main novelty of our work:

The ability to fast actuate laser frequency and low-frequency noise are of paramount importance across most of the applications of lasers. While conventional (bulk ECDL, or fiber-based) lasers have established elaborate techniques for wideband laser frequency actuation – the latter is slow. To achieve fast frequency actuation (MHz bandwidth), one typically requires external modulators, such as IQ modulators. However, the latter adds complexity and additional costly RF equipment.

We emphasize that the key novelty of our work is an LNOD-platform-based hybrid laser that **simultaneously exhibits low phase noise and ultra-fast tuning** – with 12 petahertz-per-second. The latter is a record for integrated silicon-based lasers and demonstrates a new application for lithium niobate integrated photonics. We demonstrate a flat actuation bandwidth of up to 100 MHz. Furthermore, laser self-injection locking to LNOD microresonators allows reducing the free-running laser linewidth by two orders of magnitude, yielding an intrinsic linewidth of 3 kHz.

Beyond demonstrating a new concept, we illustrate the potential of our laser source by an actual demonstration of linearization-free coherent FMCW laser ranging. We perform a 3D ranging experiment of a scene demonstrating in a system-level application the utility of our approach.

Here, we present a point-by-point reply (in **blue**) to the reviewers' comments (in **black**), as well as the action taken (in **red**) and new/rephrased sentences in the main manuscript (in **green**).

Referees' comments:

Referee #1 (Remarks to the Author):

Review of Nature 2021-10-16184

This paper reports on the hybrid integration of an injection-locked tunable laser using heterogenous integration of LiNbO₃ with silicon nitride waveguides for the tunable laser cavity and hybrid integration of a butt coupled InP gain block. The laser is then used to demonstrate fast wavelength tuning while maintain certain optical properties and application to laser ranging.

While the work is interesting and of good technical quality, it is not appropriate for publication in Nature.

The work does not represent a novel, significant breakthrough, does not report on work of outstanding scientific importance. The submitted work does not reach a conclusion of interest to the Nature interdisciplinary readership. There are major issues with the novelty and claims as well as the presentation of the results. This paper is more appropriate for another optical journal.

We disagree with the Reviewer on this conclusion. Integrated photonics based on lithium niobate is a new frontier that has brought forward new devices with improved capability in integrated photonics, as well as functionality that was not attainable before. Examples include CMOS level voltage high-rate modulators, on-chip electro-optical frequency combs

or cascaded frequency shifters (all published in Nature). Our work is novel: we report the first self-injection locked integrated photonic laser and demonstrate using this approach unprecedentedly fast tuning of petahertz-per-second tuning rate. This is a record for integrated photonics.

Moreover, we present the first-ever use of a lithium niobate-based integrated laser for FMCW LiDAR, including an actual proof of concept system demonstration. Given the wide attention that FMCW has received, we believe that our application shows the significant potential of our approach. The importance of our finding is the ability to endow integrated photonics-based lasers with both frequency agility and narrow linewidth. The work does, in our view, constitute a milestone in integrated photonics-based frequency agility – allowed by the new platform of LNOD.

In all due respect, we are surprised by the comment that our results appear not to be novel. We do faithfully report the first narrow linewidth and ultrafast tunable laser with a 12 petahertz-per-second tuning rate in integrated photonics. This is new and novel.

1. In the abstract and elsewhere, the use of “hybrid” and “heterogeneous” are not clear which can confuse readers who are not central to this area of research. The platform is a hybrid integrated laser, and as such is a very common type with butt-coupled III-IV gain element. While the tuning mechanism is heterogeneously integrated with the waveguide (lithium niobate on silicon nitride), the gain is hybrid integrated (butt coupled). There are many papers with heterogeneously integrated tuning elements. As such, the claim of a **heterogeneously integrated lithium niobate laser** and hybrid integration will be confusing to the reader and does not represent the level of novelty appropriate for a Nature publication.

a. “Laser self-injection locking is initiated by butt-coupling of an InP distributed-feedback (DFB) diode laser to the LNOD chip (Fig. 1 (c,d))”

b. “This is achieved through heterogeneous integration of ultralow-loss silicon nitride (Si₃N₄) photonic circuits with thin-film LiNbO₃ via direct wafer bonding.

c. The hybrid platform features low propagation loss of 8.5 dB/m, enabling narrow-linewidth lasing (intrinsic linewidth of 3 kHz) by self-injection locking to a III-V semiconductor laser diode.”

We respectfully disagree with the terminology comment of the Referee. We never used the phrase “heterogeneously integrated lithium niobate laser” anywhere in our manuscript, neither in the main text nor in the SI. There must be a misunderstanding.

Concerning the use of heterogeneous and hybrid integration, throughout the manuscript, we used those terms following the common convention as defined in, for example, P. Kaur et al., “Hybrid and heterogeneous photonic integration”, APL Photonics (2021):

- “**Hybrid integration** is an integration process that connects two or more PIC or photonic device chips usually from different material technologies into one single package.”
- “**Heterogeneous integration** is an integration process that combines two or more material technologies into a single PIC chip.”

A hybrid platform is also a common convention to refer to platforms fabricated by heterogeneous integration (e.g., He, M. et al. High-performance hybrid silicon and lithium niobate Mach–Zehnder modulators for 100 Gbit s⁻¹ and beyond. Nature Photonics (2019))

The described laser source uses the conventional terminology of a DFB diode laser (a gain chip) hybrid integrated to an LNOD microresonator (a microring on a passive chip). This passive chip, in turn, is fabricated by heterogeneous integration of an LNOI wafer to a Damascene Si₃N₄ wafer. The LNOD platform is a hybrid Si₃N₄/LiNbO₃ platform.

We respectfully disagree with the Referee regarding novelty evaluation. The novelty is not the LNOD platform itself but the application it enables: the petahertz-per-second tuning of a laser source with narrow linewidth. For this, however, the combined challenge of both low loss (for tight laser self-injection locking) and Pockels effect needed to be addressed. We emphasize that current state-of-the-art integrated lithium niobate-based photonic integrated circuits, except for one single report [M. Zhang et al., “Monolithic ultra-high-Q lithium niobate microring resonator”, Optica, 2017], have significant optical propagation losses. Thus, a new platform is required to achieve pronounced laser linewidth reduction for laser self-injection locking.

Our hybrid platform addresses this challenge, allowing to preserve narrow linewidth (intrinsic linewidth of 3 kHz) and unprecedentedly fast tuning (12 PHz/s). So far, similar results, however, with a 10x slower tuning rate, have been possible only with heterogeneously integrated piezoelectrical actuators made from PZT or AlN (see Ref. 13 in Table 1).

Table 1 represents a comparison of the tunable integrated laser systems and their relevant specifications. Specifically, we compare our laser performance with the lasers from other FMCW LiDAR papers with high linearity [Refs. 5, 6, 7, 8, 9, 13]. **This comparison indicates that we outperform those lasers in the tuning rate while maintaining low frequency noise.** Moreover, we do not use any OPLL, pre-distortion compensation methods or any other auxiliary procedure to achieve better linearity.

It is worth pointing out that there are two approaches demonstrating a higher **tuning rate**.

The first is Ref. [8] having a tuning rate of 1.1×10^{17} Hz/s. However, the intrinsic noise level is three orders of magnitude larger than the one demonstrated in this work. Also, the approach described there requires k-point sampling to improve linearity. Finally, in Ref. [8], a reported modulation frequency (10 kHz) is actually much smaller than the one measured in our work (10 MHz). The larger tuning rate of Ref. [8] is explained only by a wider tuning range.

The second is Ref. [12], where a lithium-niobate whispering-gallery-mode resonator is used for adiabatic frequency conversion of a laser source (examples of DFB, Ti:Sapphire, and grating-stabilized diode laser are considered). However, this approach is not an integrated photonics platform. Thus, the direct comparison is not correct. Moreover, neither linearity analysis, coherent LiDAR demonstration, nor laser frequency noise measurements are reported there.

During the review process, we became aware of a very recent competing work by the Vahala/Bowers/Lin group [16], where a tunable integrated laser source based on thin-film lithium niobate was demonstrated. Their laser, achieving higher tuning speed, has lower linewidth than reported in our work. We consider the appearance of this work during the review process of our manuscript as clear and persuasive evidence of the novelty of our work and the importance of LNOI-based integrated laser research field.

Table 1. Performance comparison of tunable laser systems.

Ref.	Active (Gain) element	Passive element	White noise floor	Modulation frequency (f_{mod})	Tuning range (B)	Tuning rate ($f_{\text{mod}} * B$) [Hz/s]	Linearity	Optical output power (mW)	FMCW LiDAR demo	Resume (comparison to this work)
1	InP RSOA	Si ₃ N ₄ chip	40 Hz	n/a (integrated heaters)	8.7 THz	n/a	n/a	23	No	No tuning rate experiments, fundamentally low tuning rate (heaters), no FMCW LiDAR demo
2	InP die	SOI chip	220 Hz	n/a (integrated heaters)	30 GHz (continuous), 13.8 THz (total)	n/a	Low	3.5	No	No tuning rate experiments, low linearity, fundamentally low tuning rate (heaters), no FMCW LiDAR demo
3	GaAs die	SOI chip	5.3 kHz	n/a (integrated heaters)	6.02 THz (total)	n/a	Low	1.5	No	No tuning rate experiments, fundamentally low tuning rate (heaters), low linearity, no FMCW LiDAR demo
4	GaAs die	SOI chip	n/a	n/a (integrated heaters)	3.1 THz	n/a	Low	2.7	No	No frequency noise measurements, no tuning rate experiments, fundamentally low tuning rate

										(heaters), low linearity, no FMCW LiDAR demo
5	DFB	SOI chip	n/a	100 Hz	64 GHz	6.4×10^{12} Hz/s	High	1	Yes	No frequency noise measurements, low tuning rate
6	DFB SCL	n/a	1 MHz	1 kHz	100 GHz	10^{14} Hz/s	High (optoelectronic feedback loop)	40	Yes	High frequency noise floor, lower tuning rate, extra linearization procedures involved
7	VCSEL	n/a	n/a	5 kHz	155 GHz	7.75×10^{14} Hz/s	High (ILC pre-distortion)	n/a	Yes	No frequency noise measurements, lower tuning rate, extra linearization procedures involved
8	VCSEL	n/a	< 1 MHz	10 kHz	11 THz	1.1×10^{17} Hz/s	High (k-point sampling)	1	Yes	High frequency noise floor, lower tuning rate, extra linearization procedures involved
9	12 stitched DFBs	n/a	< 3 MHz	330 Hz	5.56 THz	1.8×10^{15} Hz/s	High (after linearization)	12.7	Yes	High frequency noise floor, lower tuning rate, extra linearization procedures involved
10	InP RSOA	LNOI chip	n/a	n/a (integrated heaters)	6.4 THz	n/a	n/a	2.5	No	No frequency noise measurements, no tuning rate measurements, no FMCW LiDAR demo
11	InP die	LNOI chip	< 1.5 MHz	n/a (integrated electrodes)	2.8 THz	n/a	n/a	0.77	No	High frequency noise floor, no tuning rate measurements, no FMCW LiDAR demo
12	InP DFB	LNOI chip	< 1 MHz	n/a (integrated electrodes)	n/a	n/a	n/a	19.8	No	High frequency noise floor, no wavelength tuning capability, no FMCW LiDAR demo
13	ECDL	LN WGM resonator	< 1 MHz	n/a (piezo-actuator)	28 GHz	n/a	n/a	n/a	No	No frequency noise measurements, no tuning rate measurements, no FMCW LiDAR demo
14	DFB, Ti:Sa, Grating-stabilized diode laser	LN WGM resonator	n/a	40 MHz	5 GHz	40×10^{15} Hz/s [*]	n/a	n/a	No	No frequency noise measurements, no tuning rate measurements, no FMCW LiDAR demo

15	DFB	Damascene Si ₃ N ₄ chip with AlN/P/ZT	25 Hz	800 kHz	2.1 GHz	1.7×10 ¹⁵ Hz/s	High	1.5	Yes	Lower tuning rate
16	RSOA	LNOI chip	11.3 kHz	600 MHz	2 GHz	2×10 ¹⁸ Hz/s	High	< 4	No	Higher frequency noise floor, no FMCW LiDAR demo
This work	InP DFB	LNOD chip	3 kHz	10 MHz	1.2 GHz	12×10¹⁵ Hz/s	High	0.15	Yes	n/a

[1] Y. Fan, A. van Rees, P. J. Van der Slot, J. Mak, R. M. Oldenbeuving, M. Hoekman, D. Geskus, C. G. Roeloffzen, and K.-J. Boller, Hybrid integrated InP-Si₃N₄ diode laser with a 40-hz intrinsic linewidth, *Optics express* 28, 21713 (2020).

[2] M. A. Tran, D. Huang, J. Guo, T. Komljenovic, P. A. Morton, and J. E. Bowers, Ring-resonator based widely-tunable narrow-linewidth si/inp integrated lasers, *IEEE Journal of Selected Topics in Quantum Electronics* 26, 1 (2020).

[3] Aditya Malik, Joel Guo, Minh A. Tran, Geza Kurczveil, Di Liang, and John E. Bowers, "Widely tunable, heterogeneously integrated quantum-dot O-band lasers on silicon," *Photon. Res.* 8, 1551-1557 (2020)

[4] Wan, Yating, et al. "High Rate Evanescent Quantum-Dot Lasers on Si." *Laser & Photonics Reviews* 15.8 (2021): 2100057.

[5] C. V. Poulton, A. Yaacobi, D. B. Cole, M. J. Byrd, M. Raval, D. Vermeulen, and M. R. Watts, Coherent solid-state lidar with silicon photonic optical phased arrays, *Opt. Lett.* 42, 4091 (2017).

[6] N. Satyan, A. Vasilyev, G. Rakuljic, V. Leyva, and A. Yariv, Precise control of broadband frequency chirps using optoelectronic feedback, *Opt. Express* 17, 15991 (2009).

[7] X. Zhang, J. Pouls, and M. C. Wu, Laser frequency sweep linearization by iterative learning pre-distortion for fmcw lidar, *Opt. Express* 27, 9965 (2019).

[8] M. Okano and C. Chong, Swept source lidar: simultaneous fmcw ranging and nonmechanical beam steering with a wideband swept source, *Opt. Express* 28, 23898 (2020).

[9] T. Di Lazaro and G. Nehmetallah, Large-volume, low-cost, high-precision fmcw tomography using stitched dfbs, *Opt. Express* 26, 2891 (2018).

[10] Ya Han, Xian Zhang, Fujin Huang, Xiaoyue Liu, Mengyue Xu, Zhongjin Lin, Mingbo He, Siyuan Yu, Ruijun Wang, and Xinlun Cai, "Electrically pumped widely tunable O-band hybrid lithium niobate/III-V laser," *Opt. Lett.* 46, 5413-5416 (2021)

[11] Camiel Op de Beeck, Felix M. Mayor, Stijn Cuyvers, Stijn Poelman, Jason F. Herrmann, Okan Atalar, Timothy P. McKenna, Bahawal Haq, Wentao Jiang, Jeremy D. Witmer, Gunther Roelkens, Amir H. Safavi-Naeini, Raphaël Van Laer, and Bart Kuyken, "III/V-on-lithium niobate amplifiers and lasers," *Optica* 8, 1288-1289 (2021)

[12] Amirhassan Shams-Ansari, Dylan Renaud, Rebecca Cheng, Linbo Shao, Lingyan He, Di Zhu, Mengjie Yu, Hannah R. Grant, Leif Johansson, Mian Zhang, and Marko Lončar, "Electrically pumped laser transmitter integrated on thin-film lithium niobate," *Optica* 9, 408-411 (2022)

[13] Christoph S. Werner, Wataru Yoshiki, Simon J. Herr, Ingo Breunig, and Karsten Buse, "Geometric tuning: spectroscopy using whispering-gallery resonator frequency-synthesizers," *Optica* 4, 1205-1208 (2017)

[14] Yannick Minet, Luis Reis, Jan Szabados, Christoph S. Werner, Hans Zappe, Karsten Buse, and Ingo Breunig, "Pockels-effect-based adiabatic frequency conversion in ultrahigh-Q microresonators," *Opt. Express* 28, 2939-2947 (2020)

[15] Lihachev, Grigory, et al. "Ultralow-noise frequency-agile photonic integrated lasers." *arXiv preprint arXiv:2104.02990* (2021)

[16] Mingxiao Li, Lin Chang, et al. Integrated Pockels laser, arXiv:2204.12078 (2022)

[*] although it was not explicitly stated in Ref. [12], the method proposed there is potentially capable of demonstrating tuning rates of up to 200 PHz/s (for a more detailed explanation, please refer to the referee 3 reply).

Action taken: We added the comparison table of tunable integrated lasers to the SI.

2. The claim of a first lithium niobate hybrid tunable laser is not accurate. There has been prior publication of such results

a. Sun Yat-sen University in China and University of British Columbia in Canada claim the first electrically pumped hybrid lithium niobate/III-V laser [Ya Han et al, *Optics Letters*, v46, p5413, 2021].

We thank the Reviewer for pointing out this work. We note that **this work was posted online by the journal 3 days after our work was submitted** to Nature (please refer to the screenshots below). Thus, we were not aware of it. Moreover, this work did not show any continuous tuning of the laser either.

Detailed Status Information

Manuscript #	2021-10-16184
Current Revision #	0
Submission Date	8th October 21
Current Stage	Decision sent to author
Title	Ultrafast tunable lasers using lithium niobate integrated photonics
Manuscript Type	Physical Sciences - Article
	Suggested revision

YA HAN,^{1,†} XIAN ZHANG,^{1,†} FUJIN HUANG,¹ XIAOYUE LIU,¹ MINGYUE XU,¹ ZHONGJIN LIN,² MINGBO HE,² SIYUAN YU,¹ RUIJUN WANG,^{1,3} AND XINLUN CAI^{1,4}

¹State Key Laboratory of Optoelectronic Materials and Technologies, School of Electronics and Information Technology, Sun Yat-sen University, Guangzhou 510275, China

²Department of Electrical and Computer Engineering, The University of British Columbia, Vancouver, BC V6T 1Z4, Canada

³e-mail: wangrj26@mail.sysu.edu.cn

⁴e-mail: caixlun5@mail.sysu.edu.cn

Received 2 September 2021; revised 9 October 2021; accepted 11 October 2021; posted 11 October 2021 (Doc. ID 442281); published 26 October 2021

In principle, the Vernier-based scheme demonstrated in the reference could allow continuous tuning over some range; however, the implementation would be much harder, as both rings and potentially intracavity phase must be tuned synchronously with proper optical phase adjustment.

Overall, the mentioned paper does not demonstrate and impact our core novelty, petahertz per second continuous tuning with narrow linewidth and its actual application in FWC LiDAR.

Action taken: We cited this work in the abstract and introduction, despite the fact that it was published after this work was submitted to the journal.

New sentence: “Electrically pumped hybrid lithium niobate/III-V laser has been recently demonstrated using Vernier-filter-based scheme.”

3. The reported advancement is not significant enough for a Nature publication. This is acknowledged in the paper. CMOS compatible voltages would be one of the significant results needed to move this field forward.

a. with further improvements in photonic integrated circuit design, these devices can operate with CMOS-compatible voltages, or achieve millimetre-scale distance resolution.

We regret that the Reviewer finds our advancement not significant. **Our main claim is frequency agility and petahertz per second continuous tuning rate** for the hybrid laser, which to date has not been achieved in integrated silicon-based photonics [Dhoore, G. Roelkens and G. Morthier, IEEE Journal of Selected Topics in Quantum Electronics, vol. 25, no. 6, pp. 1-8, Nov.-Dec. 2019] (neither with silicon or silicon nitride - two commercial platforms available via foundry). We have attained a 100 MHz flat actuation response, a record in integrated silicon-based photonics.

Moreover, concerning the future of voltage reduction, the latter is not necessary for FMCW LiDAR and related applications such as gas sensing, where one can use a high voltage CMOS charge pump to actuate the chip with up to MHz rate.

However, as a matter of fact, our approach of a hybrid integrated lithium-niobate-based laser may also potentially achieve significantly lower voltages by increasing the participation in lithium niobate (or, for this purpose, using a fully LNOI based integrated photonics platform). This is what is actually acknowledged, and not that the novelty is not significant enough. Device performance optimization is part of future work and beyond the scope of our current study, which already demonstrates not only a new platform and integrated photonic laser concept but also an actual proof of concept FMCW ranging experiment at the system level.

4. While the claim of heterogenous integration of an electrooptically tunable material (lithium niobate) to a waveguide platform (silicon nitrate) is accurate, similar results have been published before, by multiple groups including the authors:

- a. Low-loss Thin Film Lithium Niobate Bonded on Silicon Nitride Waveguides, Siddhartha Ghosh, Siva Yegnanarayanan, Matthew Ricci, Dave Kharas, and Paul Juodawlkis, CLEO 2020.
- b. Heterogeneous integration of lithium niobate and silicon nitride waveguides for wafer-scale photonic integrated circuits on silicon, Optics Letters, Feb. 2017.

We agree with the Referee that heterogeneous integration of an electro-optically tunable material (lithium niobate) to a waveguide platform (Si_3N_4) has been demonstrated. We have also added the work that the Reviewer brought to our attention. While it is not common to cite a CLEO abstract, we have added it to the body of prior work within the abstract. We note that the work seems *not to have been published* in a peer-review paper since 2020. Nevertheless, it is now cited for completeness.

However, this work has been demonstrated on the *chiplets level* (not wafer-scale using wafer bonding). What is, however, more important are the optical losses – as these are central to maintaining noise reduction during self-injection locking. The *cited literature demonstrated no low-loss ring resonators with either wafer bonding or chiplett bonding*, which is pivotal for low noise self-injection-locked lasers.

The inferred **loss in the Optics Letters paper was 20 ± 40 dB/m**, where the error bar was larger than the estimated value and therefore does not constitute a reliable value (!). This corroborates the notion that, for low-loss measurements, it is of utmost importance to have microresonators for which quality factors can be measured. In contrast, our work demonstrates bona fide and proven **8 dB/m via resonators and via multiple measurements on a wafer that are statistically significant**.

The work by **Ghosh et al.** (“Low-loss Thin Film Lithium Niobate Bonded on Silicon Nitride Waveguides”) exhibited a **loss of 40 dB/m** – again significantly higher than our 8 dB/m (by $5\times$). Their resonators exhibit a Q-factor that is significantly lower than our work (8×10^5).

We emphasize that the only published result with a higher (3dB/m) Q-factor is the work from Loncar (see table below, Ref [17]) which is based on a *single, isolated* measurement. There is *no statistical analysis* whatsoever and the result may well be an outlier in the Q-factor distribution. In contrast, our work reports 8 dB/m – as the *most likely observed loss* based on statistical analysis.

As such, both works do not compromise the novelty of our results. The demonstrated propagation losses in the cited literature are much higher than the value demonstrated in our work, 8 dB/cm. Thus, those platforms are not suitable for high fidelity self-injection locking and low noise laser operation.

In contrast, our work uses **full wafer bonding**. Bonding at wafer level represents an advantage, as an entire 100 mm LNOI wafer is integrated onto the pre-fabricated Si_3N_4 wafer in a batch fashion, thereby increasing throughput and yield. Wafer-scale integration opens up perspectives not only for the fabrication of multiple components but also facilitates the whole fabrication, as processing at the wafer level is usually more practical and easier to implement. Chiplet bonding might be advantageous for some individual experimental devices but not for large-scale integration. Our results could also be attained using chiplets, but this would represent a lower level of development.

We put a clarifying sentence in the abstract and manuscript to emphasize the difference between chiplet and wafer bonding approaches.

We also added Table 2 in the SI to compare our results to other work on integrated lithium niobate.

Table 2. Performance comparison of LN platforms.

Ref	Intrinsic Q-factors	Linear optical loss	Statistical analysis	$\sqrt{\pi}L$ product E-O efficiency	PDK	Wafer-level fabrication
This work	4×10^6	8 dB/m	Yes	30 MHz V^{-1}	yes	yes
[17]	$\approx 10^7$	2.7 dB/m	No	not shown	no	yes
[18]	No resonators	No data	No	6.7 V cm	no	no
[19]	2.5×10^6	No data	No	500 MHz V^{-1}	no	yes

[20]	No resonators	(20±40) dB/m	No	not shown	no	no
[21]	1.8×10^6	27 dB/m	Yes	not shown	yes	yes
[22]	7.68×10^5	20 dB/m	No	5.1 V cm	no	yes
[23]	No resonators	No data	No	3.1 V cm	no	yes
[24]	No resonators	70 dB/m	No	3 V cm	no	yes
[25]	8.19×10^5	44 dB/m	Yes	not shown	no	no

- [17] M. Zhang et al., “Monolithic ultra-high- Q lithium niobate microring resonator”, Optica, 2017
- [18] N. Boynton et al., “A heterogeneously integrated silicon photonic/lithium niobate travelling wave electro-optic modulator”, Optica, 2020
- [19] M. Zhang et al., “Electronically programmable photonic molecule”, Nature Photonics, 2019
- [20] L. Chang et al., “Heterogeneous integration of lithium niobate and silicon nitride waveguides for wafer-scale photonic integrated circuits on silicon”, Optics Express, 2018
- [21] K. Luke et al., “Wafer-scale low-loss lithium niobate photonic integrated circuits”, Optics Express, 2020
- [22] A. Ahmed et al., “High-performance race track resonator in silicon nitride – thin film lithium niobate hybrid platform”, Optics Express 28, 1868 (2019)
- [23] A. Rao et al., “High-performance and linear thin-film lithium niobate Mach-Zehnder modulators on silicon up to 50GHz”, Optics Letters, 41, 5700 (2016)
- [24] S. Jin et al., “LiNbO₃ thin-film modulators using silicon nitride surface ridge waveguides”, IEEE Photonics Technology Letters 28, 736 (2016)
- [25] Ghosh, Siddhartha, et al. "Low-loss Thin Film Lithium Niobate Bonded on Silicon Nitride Waveguides." *CLEO: Science and Innovations*. Optical Society of America, 2020.

Action taken: We emphasized the difference between chiplet and wafer bonding approaches. We added a table to the SI to compare with other integrated lithium niobate platforms. We also cite Ref. [22] in the abstract.

New sentence: “This is achieved through integration of ultra-low-loss Si₃N₄ photonic circuits with thin-film LiNbO₃ via direct bonding at wafer level, in contrast to previously demonstrated chiplet-level integration”.

5. The waveguide loss and laser performance do not represent a significant breakthrough. There have been multiple publications of injection-locked hybrid integrated lasers, and even of heterogeneous integrated injection-locked lasers, as well as fast tuning in a heterogenous platform:

Before discussing the laser performance, we would like to discuss the reported waveguide loss of the hybrid platform. While numerous papers claim that the LNOI platform has reached the level of 3 dB/m loss, there is, to date, **only a single report** since 2017 on a high Q (10 million quality factor) resonator in lithium niobate **with no statistical analysis** by the M. Loncar group [M. Zhang et al., Optica, 2017].

We achieve a quality factor of more than 2.5 million over an entire wafer. **We characterize the wafer using statistical analysis for many resonances** of several chips from different wafer fields, and not only for a single resonator as was reported in Ref. [17].

Moreover, even the latest results from the group of M. Loncar (Hyperlight company) exhibit loss significantly above what we report. We show their results in Table 2 [K. Luke et al. 2020]: average losses of 27 dB/m (reaching a maximum value of 36 dB/m), significantly higher than the 8.5 dB/m that we reported here.

The increased loss makes a significant difference in the performance of laser self-injection locking, as it influences the laser linewidth in an inverse quadratic fashion. Indeed, it is not possible to achieve results similar to ours in terms of laser frequency noise suppression without maintaining low losses.

We perform wafer-scale characterization across the full wafer of microresonators (21 GHz FSR) to confirm the reproducibility of the resonances' high-quality factor. Figure 1 shows a wafer map with average linewidths (each field is one chip).

Figure 1. Wafer map with averaged linewidth indicated for the microresonators with a FSR of 21 GHz.

a. Hertz-linewidth semiconductor lasers using CMOS-ready ultra-high-Q microresonators, Warren Jin, Qi-Fan Yang, Lin Chang, Boqiang Shen, Heming Wang, Mark A. Leal, Lue Wu, Maodong Gao, Avi Feshali, Mario Paniccia, Kerry J. Vahala & John E. Bowers, **Nature Photonics** volume 15, pages 346–353 (2021)

We are ourselves inspired by this work and the more recent work of Prof. Bowers [doi: 10.1364/OL.439720]. In fact, in our lab, we recently achieved a phase noise level better than an ECDL and on par with a fiber laser using a tightly confining Si₃N₄ platform [arxiv:2104.02990]. The interest in hybrid integrated self-injection-locked lasers is currently huge, with the Jin et al., Nature Photonics paper receiving more than 85 citations in one year. However, the direct comparison of these results to our work is not suitable for the following reasons.

Firstly, the laser reported by the Vahala and Bowers groups is a fixed-frequency laser and has no element for continuous mode-hop-free tuning. Secondly, the laser uses a highly specialized platform: low-confinement, 100 nm thin Si₃N₄ waveguides. This makes the die size very large, 20 mm × 33 mm (with a microring bending radius of ~ 1 mm), which implies that the entire DUV reticle can only be used for one laser, and thus a 200 mm wafer can only produce ~40 lasers. In contrast, our material system produces tight confinement of the light, and thus the chips are only 5 mm x 5 mm (with a microring bending radius of 0.25 mm). Thirdly, our platform allows for the integration of lithium niobate, which is not possible for the low confinement platform due to the low effective refractive index of the waveguide mode. Lithium niobate bonding would not be possible, as it would lead to major losses in the case of thin top cladding (2 μm) or extremely inefficient tuning due to low optical mode overlap with lithium niobate in case of thick top cladding (14.5 μm). Our system thus enables ultrafast laser tuning that would not be possible to demonstrate in prior work.

In addition, the cited paper also does not show a realistic application scenario. In our work, we show a bonafide coherent ranging experiment without any additional linearization procedures due to excellent tuning linearity. Our extensive measurements prove the viability of our approach to fast mode-hop-free tuning in an actual FCMW laser demonstration.

Finally, we note that the foundation underlying both the Vahala/Bowers paper and our paper has been known for a long time. The company OEwaves has been using laser self-injection locking since 2003. Using bulk crystalline WGM resonators with $Q > 10^9$, an integral linewidth of 30 Hz was demonstrated.

b. Fully Integrated Photonic Millimeter-Wave Tracking Generators on the Heterogeneous III–V/Si Platform Rui-Lin Chao, Linjun Liang, Jin-Wei Shi, Tin Komljenovic, Jared Hulme, M. J. Kennedy, and J. E. Bowers

This work demonstrates a totally different approach using two DFBs and a complicated external OPLL scheme, so the performance cannot be compared. As of today, an OPLL cannot be implemented on a single chip together with the laser. Our approach does not require an OPLL. Moreover, in this paper, the intrinsic linewidth is only reduced from 2 MHz to 200 kHz, which is still much higher than in our approach.

Action taken: We added citations to the mentioned work.

6. The performance in terms of optical waveguide loss, resonator Q, and intrinsic linewidth do not represent record values in integrated photonics.

a. “The hybrid platform features low propagation loss of 8.5 dB/m, enabling narrow-linewidth lasing (intrinsic linewidth of 3 kHz) by self-injection locking to a III-V semiconductor laser diode.”

We respectfully disagree, as one needs to compare values of loss for each platform.

With respect to **waveguide loss**, for an electro-optical platform, **we have achieved exceptionally low loss, surpassed only by one single report**, which is an **isolated measurement of a *single* resonance** that has not been reproduced by others for several years (reference 1 in Table 2).

As for the Q of the resonator, our value of 4.6×10^6 **for a lithium niobate device is an excellent value, on par with or better than most pure Si₃N₄ integrated photonics circuits**, a significantly lower loss than silicon photonics, and surpassed only by platforms with very low confinement [Jin et al. Nat. Phot. 2021], our own Damascene platform [Liu, J. et al., Nat. Commun. 12, 2236 (2021)], or wide over-moded Si₃N₄ waveguides [Ji, X., Jang, et al., Laser & Photonics Reviews 2020, 15, 2000353].

For the intrinsic laser linewidth, please refer to Table 1, comparing our results to other integrated lasers. Note that neither other material platform has ever demonstrated similar or better performance, nor lithium-niobate-based one.

We would like to highlight that **the main novelty of our work is the ultrafast tuning combined with the narrow linewidth of the laser**, which is a major requirement for FMCW LIDAR. As such, our results constitute a record in terms of phase noise for an integrated laser source, generally, and for a lithium-niobate-based one, specifically.

Referee #2 (Remarks to the Author):

A. Summary of the key results

The work submitted for review presents the performance of a hybrid integrated laser source realized combining a Distributed Feedback Grating (DFB) laser in InP with a ring resonator defined in the Lithium Niobate on Damascene Si₃N₄ (LNOD) platform. The external ring resonator cavity can potentially have extremely high quality factor and provides a weak back-reflected wave to the DFB, causing injection locking regime in specific conditions. The laser is tuned in locking regime by changing the resonance frequency of the ring resonator by the electro-optic Pockel effect. The authors claim to reach an ultra-fast tuning rate of 12 PHz/s over a range of 600 MHz, while keeping a laser linewidth of 100 MHz. The laser swept source is then deployed on a coherent LIDAR experiment using a Frequency-Modulated Continuous Wave (FMCW) scheme, demonstrating a resolution of 15 cm in a 3 m range.

B. Originality and significance: if not novel, please include reference

The main novelty, according to the authors, lies in the realization of integrated laser using their LNOD platform. Their platform allows to combine high-confinement silicon-nitride waveguides with the electro-optic material lithium niobate, so to obtain compact, high-quality factor and ultrafast tuning. While the concept is interesting, it is very clear that there is large room of improvement: currently, the performance of such laser does not reach what has been already reported in literature (see Jin, W., Yang, QF., Chang, L. et al. Hertz-linewidth semiconductor lasers using CMOS-ready ultra-high-Q microresonators. Nat. Photonics 15, 346–353 (2021). <https://doi.org/10.1038/s41566-021-00761-7>).

As the same point has been raised and addressed already in reply to Referee #1 comments, we kindly ask the reader to refer to p. 9.

Action taken: We added a citation to mentioned work in the main text.

New sentence: “This technique, in conjunction with low-loss photonic integrated microresonators, has already enabled compact chip-scale lasers with sub-Hz Lorentzian linewidth using Si₃N₄”.

The main result, being the extremely fast tuning rate, is not enough when the laser linewidth is 100 MHz and the tuning range is limited to 600 MHz, which provides poor resolution in a FMCW LIDAR scheme. Moreover, the deviation is far too high (reaching 100 MHz) to be acceptable in such a narrow range.

It appears that the Reviewer has misread the manuscript. Our integrated linewidth is not 100 MHz **but rather 56 kHz at 0.1 ms integration time. The intrinsic linewidth is 3 kHz**. Such laser frequency noise performance is sufficient to perform coherent ranging at distances < 100 m.

Optical frequency excursion (B) of the laser determines the FMCW LiDAR distance resolution, which is given by $c/2B$, where c is the rate of light. For our laser, the optical frequency excursion is 600 MHz and can eventually be increased by increasing the laser locking range, for example, by introducing a loop mirror in the drop-port of the LNOD microresonator.

A large deviation from the linear ramp is only observed for the fast tuning rate of 10 MHz, which we partially attribute to the limited bandwidth of our signal generator. A 10 MHz tuning rate is beyond the range relevant to real FMCW LiDAR, but it does constitute a record tuning rate for continuous tuning with narrow linewidth.

C. Data & methodology: validity of approach, quality of data, quality of presentation

The work is well presented, and data are very well represented. All results are exhaustively reported, and the readers have all the elements to draw their conclusions. The paper follows a clear a logical structure. However, there is little comparison with state-of-the-art works.

We thank the Reviewer for the positive evaluation of the data presentation. To improve comparison with the state-of-the-art, we added two tables to the SI (see Table 1 and Table 2 of this reply). We compare our results to other lithium niobate platforms and integrated lasers.

Action taken: We added comparison to state of the art in Table 1 and Table 2 in the SI.

There are notable errors, for example in line 103, there is a contradiction between the Fig. 2d for the direction of back-reflected radiation (counter-clockwise against clockwise). Also, some information is missing: the description of the LNOD platform does not include all the thicknesses of the various layers; the properties of the DFB laser source seed are not clearly stated in the text (RIN, LI curve, grating strength and so on) and they are only reported for a single operating point in Fig. 2c. Also, there is no indication on the output power of the hybrid laser.

We thank the Referee for pointing out these omissions and errors.

We changed counter-clockwise to clockwise in the main text.

Figure 1(b) shows an SEM cross-section of the LNOD stack with a 1 μm scalebar. We added the precise thickness of the layers to the main manuscript: bottom silica cladding - 4 μm , Si_3N_4 - 950 nm, silica top spacer - 150 nm, lithium niobate - 300 nm, metal electrodes - 200 nm.

We added characterization data for the free-running DFB source to the SI.

The hybrid laser output power is given in Table 1 added to the SI.

Figure 2. Characterization of the free-running DFB source. a) Laser diode free-space optical power vs. diode current. b) Free-running DFB optical spectra at various driving currents.

Figure 3. Free running DFB RIN

Action taken: We changed counter-clockwise to clockwise in the main manuscript. We provided the thickness of the various device layers in the main text. We added the following free-running DFB characterization data to the SI: optical output power vs. diode current and the optical spectrum of the free-running laser at various current levels.

D. Appropriate use of statistics and treatment of uncertainties

The experimental data are represented together with their deviation from the intended value. Such data give clear indication of the performance of the device and the range of applicability. However, since the device relies heavily on a structure realised using a custom fabrication process, one should expect a better treatment of the statistics on the ring resonator properties.

We use the same chip characterization methods as in most of our previous papers. All the devices that were used in this study have been characterized with a state-of-the-art spectroscopy technique employing a frequency-comb-calibrated diode laser over a spectral bandwidth that covers the S, C, and L bands of the optical communication spectrum. We provide broadband measurements of the transmission and reflection spectra and detailed results for transmission and coupling for each resonance in the band of interest. The statistical method of loss characterization established in our earlier works has become the standard in the field of integrated photonic microresonators.

Action taken: We added characterization data for 4 microresonators of a LNOD wafer to the SI (see Fig. 6 of this reply). We stated the cavity linewidth and the loaded Q factor of the resonators. Furthermore, we added the reference to our previous paper, where we described our characterization setup (frequency-comb-assisted laser spectroscopy of chips) and methods in great detail.

It is apparent that both the quality factor and the reflection coefficient should be much higher for better performance. Moreover, it is expected that, with the improvement of the fabrication process, the quality factor will be enhanced whereas the back-reflection coefficient, which is triggered by scattering processes introduced by imperfections, will be minimised. What is the trade-off?

We agree with the Reviewer that a higher quality factor will lead to better frequency noise suppression. In a theoretical model based on only white noise contribution, the suppression ratio scales as $Q^2_{\text{microring}}$. Higher backreflection increases locking range and does not significantly change frequency noise suppression. In fact, it is not true that the backscattering is smaller for higher Q devices. Crystalline WGM microresonators from MgF_2 with $Q_0 > 10^9$ exhibit on-resonance back reflection up to 15% [Pavlov N. et al., Nat Phot. 2019]. In integrated Si_3N_4 microresonators, we experimentally observed that lower FSR devices have smaller backreflection [see arxiv:2104.02990, SI Figure 5], contrary to Rayleigh scattering theory [M.L. Gorodetsky, et al. J. Opt. Soc. Am. B 17, 1051-1057 (2000)].

An important trade-off is between coupling to the cavity and the output laser power. Our microresonator is almost critically coupled. Thus, in the SIL regime (locked to a cavity resonance), the transmission and corresponding laser output power is small (150 uW in our case). Use of overcoupled microresonators and/or on-chip splitters that return a small fraction of the light for injection locking would increase the output power.

Theoretical analysis and possible optimizations for laser self-injection locking to a high-Q microresonator are presented in [arxiv:2106.15237]. We added a discussion of laser self-injection locking optimization to the main text.

Action taken: We cite [Galiev et al. 2021] and add a discussion of laser self-injection locking optimization to the main text.

New sentence: “Laser stabilization for any level of intrinsic backscattering can be further improved by introducing an on-chip drop-port-coupled loop-mirror. The locking range and frequency noise suppression can be further improved by tuning the coupling and feedback phase of the drop-port mirror.”

Furthermore, there is no mention to the reflection introduced by the butt-coupling and the end-facet on the other side, which inevitably play a role, unless some actions were taken.

Si₃N₄ chip facets, separated by 4.96 mm, form a Fabry-Perot cavity with a ~4% extinction ratio (see Figure 2(a) of the main manuscript). This FP cavity has low finesse (~2) and does not lead to laser self-injection locking because the optical feedback is very broadband. Nevertheless, we agree with the Reviewer that additional action might be taken to reduce reflection from the chip facets, such as using angled output tapers.

Action taken: We added a sentence to the main text indicating how reflections from the chip facets may be mitigated.

New sentence: “The reflection from the chip facets can be reduced by using angled output tapers.”

Finally, the most important result of this work, being the tuning rate at 12 PHz/s, is measured over a range affected by a deviation that could be unacceptable in most applications.

We agree on the novelty of the result but disagree on the assessment of our data. A large deviation from the linear ramp is only observed for the fast tuning rate of 10 MHz (Figure 3 (c)). We attribute it to the limited bandwidth of our ramp signal generator. Moreover, the 10 MHz tuning rate is not relevant to real FMCW LiDAR as most commercial solutions use at most 100 kHz ramp frequency. Utilizing 10 MHz ramp frequency for coherent ranging will lead to shorter measurement time and lower SNR on the photodetector in the delayed homodyne scheme. In addition, the beatnote signal for the 3 m target will be in the GHz range, which requires high-frequency photodetectors and electronics to capture and process the data, and currently is not commercially practical.

The performance and operating range of the hybrid laser heavily relies on the fabrication process, which at the current stage, has not developed the required maturity yet to stand-out and make a breakthrough in the realization of optical sources with superior performance.

We respectfully disagree. Thin-film Si₃N₄ has been a commercial technology provided by foundries for several years now. Processing of Si₃N₄ photonics is now possible on 8-inch wafers (e.g. from XFAB/Ligentec and IMEC). LNOI is also commercially available (from NanoLN) and is broadly used in many labs and companies, and the wafer bonding process can be purchased from various companies (e.g., Thorlabs CMS). Our technology is not unlike other integrated platforms, such as silicon photonics, in its reliance on numerous materials and processes. Although relatively new technology, we believe all the ingredients of our platform have reached the required maturity to be in the spotlight for the photonics community, as is evident from the current enormous interest in such an integrated photonics platform by the research community and industry.

It is not clear what is the level of control on the properties of the fabricated ring resonator, and the whole device.

Fabrication of Si₃N₄ integrated photonic circuits using either a subtractive or Damascene process is available at the level of a process design kit (PDK) from XFAB/Ligentec or IMEC. Waveguide height can be controlled to a precision of ~20 nm.

We previously reported high-yield, wafer-scale fabrication of ultra-low-loss silicon nitride photonics in our group [Liu, J., et al. Nat Commun 12, 2236 (2021), specifically Supplementary Note 3]. Figure 4 shows that we can achieve a high level of control on the properties of fabricated microrings. Here, resonators have only 10 GHz FSR and uniform Q over fields in the center of the wafer.

Figure 4. Data from [Liu, J., et al. Nat. Commun. 12, 2236 (2021)]. Loss distribution on a 4-inch wafer consisting of chips with 10-GHz-FSR resonators. (a) Photo of the 4-inch wafer. (b) Layout for DUV stepper lithography, with the most commonly observed value for the intrinsic cavity linewidth on chip C15. (c) Chip design.

Lithium niobate bonding was reproduced several times for LNOI wafers with diameters of 2 inches, 3 inches, and 4 inches (see Fig. 5).

Figure 5. Photos of LNOD wafers after bonding and removal of the LNOI silicon substrate for 3-inch (left) and 4-inch (right) LNOI wafers.

Action taken: We added a citation to the high-yield photonic Damascene fabrication process [Liu, J., et al. Nat. Commun. 12, 2236 (2021)]. We added characterization data for other devices from the LNOD wafer to the SI.

Furthermore, the performance depends also on the properties of the **DFB laser source, for which is unknown the reproducibility**. It is also not clear what are the advantages for this device offered by the Damascene process for the definition of the silicon nitride waveguides, which gives the name to the platform.

Nowadays, DFB lasers are commercially available from many companies. Even high-power DFBs with >100 mW output power are readily available, e.g., from Freedom Photonics or PhotonX. The fabrication of these DFBs is quite reproducible. We agree that DFB phase noise and RIN can vary, as we observed in our experiments with different lasers. However, laser self-injection locking is universally manifested in frequency noise suppression for any type of laser used. The main factor is narrowband frequency-selective optical feedback from the resonator. Laser self-injection locking has been demonstrated with high-Q Fabry-Perot (FP) cavities, WGM bulk crystalline microresonators and integrated photonic resonators using FP multifrequency, DFB and VCSEL lasers at wavelengths from near the UV to the near IR. Our work adds continuous tuning capability to self-injection-locked lasers via the Pockels effect in the bonded lithium niobate layer. The properties of the Si₃N₄ photonic integrated circuit (PIC) dictate the final performance of the laser: input coupling, loaded cavity linewidth and backreflection coefficient are the key factors. The Damascene process that we use for fabrication of our Si₃N₄ PICs has a clear advantage over the conventional subtractive process, as it allows easy planarization of the top cladding via CMP to achieve the required roughness for subsequent wafer bonding. In the subtractive process, it is probably not impossible to do the same, but the increased topography of the nitride waveguide will require a more involved planarization process employing a thick top cladding.

F. Suggested improvements: experiments, data for possible revision

At the current stage, it would be important to have statistical data on the ring resonator, and different devices realized with different rings.

We thank the Reviewer for the suggested improvements. We have implemented them in this reply and in the revised manuscript. Chip characterization data across the wafer are added to the SI. Data includes the intrinsic linewidths and coupling strengths for different chips from the same wafer. We use a state-of-the-art spectroscopy technique employing a frequency-comb-calibrated diode laser over a spectral bandwidth covering the S, C, and L bands, the same characterization method as used for most of our previous papers.

Figure 6 presents new characterization data for two fields of the D67 LNOD wafer. The median intrinsic cavity linewidth is in the 50–65 MHz range.

Figure 6. (a,d,g,j) Transmission (blue) of the bus waveguide coupled to an LNOD microresonator with a free spectral range (FSR) of 102 GHz (C11) from 3 fields of the wafer and different microresonators on a chip (F7 WG3.3, F1 WG3.3, F7 WG4.2, F5 WG4.1); (b,e,h,k) Frequency-dependent microresonator loss $\kappa_0/2\pi$ (green) and bus waveguide coupling $\kappa_{ex}/2\pi$ (blue). (c,f,i,l) Histogram of intrinsic microresonator loss rate.

We also provide a C11 chip design for reference in Figure 7. WG4.1 means microresonator in 4th row, 1st from the left.

Figure 7. Design of C11 chip with LNOD microresonators having a FSR of 102 GHz.

Action taken: We added new characterization data for the LNOD D67 wafer to the SI.

It would be also useful to have the step response of the device to understand the dynamic of the tuning.

To address the Reviewer's comment, we performed step response measurements. We tuned the reference laser (Toptica CTL) to the slope of a cavity resonance. We then applied a square signal to the electrodes and measured the cavity transmission. Figure 8 shows the measurement results for a repetition rate of 1 kHz. Fit of the step up (Figure 8 (c) and the step down (Figure 8 (d) of the cavity transmission give a rise time of 5 ns and a fall time of 9 ns, respectively.

Figure 8. Step response of the LNOD actuator. (a) Input signal from the signal generator. (b) Cavity transmission. (c) Leading edge of transmission step. (d) Falling edge of transmission step.

G. References: appropriate credit to previous work?

The work provides appropriate credit to previous work for all the concepts described throughout.

We thank the Reviewer for this acknowledgment.

H. Clarity and context: lucidity of abstract/summary, appropriateness of abstract, introduction and conclusions

The results are reported in various points of the manuscript, abstract, introduction and conclusions. The abstract however already includes possible future works and improvements that should be exclusive of the conclusions only. In the introduction, final results are included and they can confuse the reader.

We thank the Reviewer for pointing out these deficiencies in the abstract and introduction and have addressed them.

Action taken: We moved the discussion of CMOS-compatible voltages from the abstract to the conclusion. We removed the final results (achieved laser metrics) from the introduction.

Referee #3 (Remarks to the Author):

The authors report on self-injection locking of a DFB laser by an electro-optically tunable micro-resonator based on a lithium-niobate-on-insulator ring cavity. Tuning for about 600 MHz is demonstrated with rates of up to 10 MHz. The system is tried and tested in a frequency-modulated continuous-wave radar system, providing 16 cm depth resolution for a target in 3 m distance.

There is a wealth of applications for tunable lasers with electrical wavelength control. I am confident that laser systems like those demonstrated here will become relevant. The implementation shown here is new, and the experimental work appears to be solid and profound. Thus I am in favor to accept this manuscript for publication in Nature Photonics.

We thank Reviewer #3 for their careful study of our manuscript, constructive suggestions, and their positive assessment of the novelty and impact of our manuscript. We are happy to provide below a detailed response to the questions raised.

There are, however, some deficiencies that should be resolved before publishing this work. Below I will list my points in the order of occurrence, but let me start with some general statements:

Some relevant citations are missing, being essential to put the work into the right context and to give appropriate credit to previous work.

Some parts of the descriptions are incomplete.

Although I share the enthusiasm for this innovation, one should be realistic which applications such lasers can address and which not. Hence some exaggerations need to be removed and a more realistic discussion about the pending issues would be appreciated.

We thank the Reviewer for pointing out these deficiencies. We modified the abstract and conclusion sections accordingly.

Line 35: Please do not mention OCT. To my mind it is not feasible that this light source will be applied there. For OCT tuning by hundreds or thousands of GHz is needed, being out of range for the method introduced here. Furthermore, there are established solutions for OCT; there is not much pressure for better lasers in this area.

We agree with the Reviewer that achieving mode-hop-free tuning over tens of nanometers is challenging for our laser.

Action taken: We removed the mention of OCT applications from the manuscript.

Line 40: Here it is stated that the first integrated photonics based on lithium niobate has been realized just recently. This is not true and should be corrected. Please see M. A. Abore und M. M. Fejer, Opt. Lett. 22, 151 (1997), G. Schreiber et al., Proc. SPIE 4277, 144 (2001) for an early review and Guarino et al., Nat. Photon. 1, 407 (2007), just as some examples. I would also call the standard telecom modulators, that are based on waveguides in LiNbO₃, already an integrated photonic solution. Thus there exists already a mass application.

We thank the Reviewer for pointing out good papers on early lithium niobate work. We have added citations to both papers to the introduction. However, it is debatable whether standard telecom lithium niobate modulators can be called integrated photonic solutions, and such a description is not widely accepted in the community of integrated photonics. Put simply, standard telecom lithium niobate modulators do not use the advanced wafer-scale processing developed in the microelectronics industry, such as etching, lithography, etc., to produce miniaturized, highly integrated devices. In contrast, optical waveguides in most commercial devices are defined by ion diffusion or proton exchange methods, resulting in low index contrast and weak optical mode confinement. As accepted by the community, an integrated photonic solution should feature sub-wavelength-scale light confinement and dense integration of optical and electrical components, which is not the case for conventional lithium niobate devices.

Action taken: We added references to the mentioned review in the abstract.

Line 44: Optional, with regard to integrated high-Q LNOI resonators – Referencing Wolf et al., Optics Express 25, 29927 (2017)

We thank the Reviewer for pointing out this reference, showing batch polishing of etched lithium niobate waveguides.

Action taken: We added reference to this work in the main text.

Line 59: Optional, with regard to piezoelectric tuning – Referencing Ch. Werner et al., Optica 4, 1205-1208 (2017) and compare the specs to those of the work under consideration.

We thank the Reviewer for pointing out this reference. We added this work to Table 1 in the SI comparing tunable laser systems.

Action taken: We added a citation to this work. We also added it to Table 1, comparing tunable lasers.

Line 59: The closest technical competition to the system presented in this work is the adiabatic electro-optic tuning of laser light with LiNbO₃ micro-resonators, please see Minet et al., Opt. Express 28, 2939 (2020). The tuning rate achieved there is 250 PHz/s, instead of 12 PHz/s, as presented in the work under consideration.

We thank the Reviewer for pointing out this reference. This remarkable work shows adiabatic frequency conversion in a WGM crystalline resonator made of lithium niobate. Tuning of 266 MHz/V has been demonstrated for e-polarized waves. However, no laser self-injection locking and laser frequency tuning have been shown in this work.

Regarding the value of the tuning rate, our evaluations provide a different, yet close enough, estimate of the tuning rate suggested by the Referee. If we take the data from Fig. 5 of the mentioned manuscript, we observe 1 GHz of frequency tuning within 25 ns of time span. From these numbers, we derive 40 PHz/s of tuning rate. If we then refer to Fig. 4, we will infer that the largest frequency excursion reported there is 5 GHz, which would potentially give 200 PHz/s of tuning rate.

Action taken: We added a citation to this work to the main manuscript.

Line 89: What is the purpose of the Al layer? If needed for a stable process: Is there an impact on the optical transparency stemming from an Al-mediated electrical conductivity?

There is no mention of an Al layer in line 89; it is Al₂O₃ that was actually used: “Next, a few-nanometer-thick **alumina layer** is deposited by atomic layer deposition (ALD) on both the donor (LNOI) and the acceptor (Damascene) wafers before contact bonding and donor-wafer removal”.

Al₂O₃ is a dielectric material and has no significant influence on optical transparency. On the ALD step, hydrogen is present, and during the bonding, water is created on the interface. All this could create losses; however, no noticeable degradation of Q-factors (that are still on the order of several million) was eventually observed.

Line 90: The process called “donor-wafer removal” should be explained more thoroughly, maybe in two sentences. What is the yield of the (bonding) process?

The process is described in the Device fabrication subsection of the Methods section: “The silicon on the backside of the donor wafer is ground, and the residual silicon after grinding is removed with tetramethylammonium hydroxide (TMAH) wet etching. The thermal SiO₂ is wet etched with buffered hydrofluoric acid (BHF).”

The bonding yield is nearly 100%, as is from the images in Figure 9 of the bonded wafer after removal of the donor wafer. Here 4-inch LNOI wafers were bonded to 4-inch Damascene silicon nitride wafers.

We fabricated two wafers for this project and three additional wafers after the submission of the manuscript. In all cases, the bonding yield was close to 100%.

Figure 9. Photographs of two different LNOD wafers after successful bonding.

Action taken: We added discussion on the bonding process yield to the main manuscript.

New sentence: “Wafer bonding yield is close to 100%. We successfully bonded five out of five wafers during three different fabrication runs.”

Line 119: Since there is no way to vary the coupling strength for a device given, where do you know from that you are close to critical coupling?

We learn that from our characterization method, where we perform a broadband scan of two lasers (using stitching of transmission traces in overlapping regions), covering the 1490-1630 nm wavelength range. This allows us to fit many resonances and study frequency-dependent bus waveguide-ring coupling. As the extrinsic coupling rate is supposed to grow with the increase in wavelength, one can easily distinguish between it and the intrinsic coupling rate that does not have any trend with a change of wavelength. By plotting the fitted intrinsic and extrinsic decay rates, one observes their relation to infer if a resonator is critically coupled or over/under-coupled. For example, Figure 10 shows under-coupled devices.

Figure 10. Frequency-dependent microresonator loss $\kappa_0/2\pi$ (green) and bus waveguide coupling $\kappa_{ex}/2\pi$ (blue).

Action taken: We added the description of our characterization method and setup, along with the extensive coupling characterization data, to the SI with the references.

Line 147/148: The Toptica laser is sufficiently stable to catch via the heterodyne measurements the stability of the realized, integrated laser? Or has the Menlo laser, mentioned in line 248, been used as a reference?

Yes, the Toptica CTL laser has sufficiently low frequency noise at offsets above 1 kHz to use it as a reference laser for heterodyne measurements with our SIL laser (see Figure 11 below). We used a Menlo Systems ORS 1 Hz laser to measure the frequency noise of the Toptica CTL.

Figure 11. Frequency noise of the reference Toptica CTL laser

Line 178: Please write which optical polarization the different beams have and what happens with the polarization in the circuit and which impact this has onto the beat signal. Mixed polarizations will fade the signal of coherent detection.

We thank the Reviewer for this comment. We use TE mode of integrated silicon nitride optical waveguide. In our experiment, we always adjust polarization with a fiber polarization controller in the reference arm of the delayed self-homodyne setup to maximize the beat note signal SNR on the balanced PD. We added the FPC element in schematics of the experimental setup in Figure 4(a) in the main text for clarity.

Action taken: We added the FPC to the schematic of the experimental setup in Figure 4(a) of the manuscript. We also added a sentence describing the use of the polarization controller in one arm of the delayed self-homodyne setup to the main text.

New sentence: “We adjust optical polarization with a fiber polarization controller in the reference arm of the delayed self-homodyne setup to maximize beatnote signal SNR.”

Line 273: The optical power after (!) the EDFA is just 4 mW. How much is it before? Please add this information to the text. Please consider in the discussion: Optical amplifiers are available just for some few spectral domains. In case that the light source without amplification delivers below 1 mW, most of the applications mentioned in concluding part of the article cannot be realized. Hence the question arises: How can the power be enhanced, without using subsequent amplification? What are the power limits?

We thank the Reviewer for this comment. Indeed, the output power of our laser in an injection-locked state is just 150 uW. The main reason is that we used an almost critically coupled microresonator, as shown in the characterization data (see Figure 6). In the SIL state, on-resonance transmitted power (laser output) is low. However, the output power can be easily enhanced by using overcoupled microresonators. As indicated above in response to Referee #2, another option is to use an on-chip splitter. Specifically, after the splitter, 10% of the DFB power is sent to the LNOD microresonator for self-injection locking, while the remaining 90% of the power goes to the output port. Such a configuration is used in commercial lasers from OEwaves, where they use free-space optics (beamsplitter, prism coupler, lenses) in hybrid packaging of high-Q crystalline microresonators and diode lasers.

Action taken: We explicitly mention the output power of 150 uW of our laser in the Methods section.

Fig. 2e: Where does the “+ 3 GHz” come from?

This measurement was performed by beating the tunable integrated laser with the output of the reference Toptica CTL laser. The heterodyne beatnote frequency was 3 GHz. We accordingly shifted the data by 3 GHz offset for plotting.

Fig. 2g: The laser frequency follows the applied field only approximately in a linear manner and this also just in a limited range. Any idea where this comes from? Linearity is crucial for most applications, FMCW LiDAR in particular.

We agree with the Reviewer that linearity is crucial for the FMCW LiDAR application. To improve the consistency of the paper, we added a citation [Behroozpour et al. (2017)]. This paper constitutes an excellent tutorial on FMCW LiDAR with an analysis of most requirements, including laser noise and tuning linearity.

Indeed, Figure 2 (g) shows that linearity is preserved only in the center part of the locking range. That is why, in the FMCW LiDAR demo, we used a reduced tuning range of 600 MHz to maintain good linearity. In addition to the flat actuation bandwidth of the LNOD device (as shown in Figure 3a), tight laser self-injection locking is a must to preserve good linearity of tuning. In our case, the laser frequency does not follow the shifting resonance precisely due to a rather loose locking. The locking point could be further optimized by carefully adjusting the optical feedback phase (or packaging). Theoretical analysis and possible optimizations for laser self-injection locking to a high-Q microresonator are presented in [arxiv:2106.15237].

Action taken: We added a citation to [Behroozpour et al. (2017)] to the main manuscript to support our statement of the benefits of low noise lasers for FMCW LiDAR.

New sentence: “Laser phase noise limits the maximum operating distance and ranging precision in this method. However, a key requirement for FMCW LiDAR at long range is frequency agility, i.e., fast, linear and hysteresis-free tuning.”

Fig. 3c: From this panel I get the impression, that the best tuning is achieved at 100 kHz, which is surprising. I would suppose that slower is better. Is there an explanation for this?

Please refer to our reply to the previous comment. Due to non-optimal parameters affecting injection locking (backreflection and optical feedback phase), the performance is slightly worse in Figure 3(c) for 1 and 10 kHz. In contrast, for the 100 kHz tuning rate, which we used in the LiDAR demo and which is used in commercial LiDAR systems, we performed the most careful adjustment of the locking point by adjusting the optical feedback phase and driving voltage amplitude. We monitor the homodyne beatnote from a reference fiber MZI and minimize the FWHM of the beat note signal to reach an optimal SIL regime.

Fig. 4b: Which times are specified in the inbox?

We thank the Reviewer for this remark. Figure 4(b) shows three examples of homodyne beatnotes from a target during the laser beam scan. The time values correspond to different timeslices (sliced in 128k bins) of the full time-frequency map presented in Figure 6 of the SI.

Action taken: We amended the legend and caption of Figure 4(b).

Karsten Buse

Reviewer Reports on the First Revision:

Referees' comments:

Referee #1 (Remarks to the Author):

Review of Nature 2021-10-16184A-Z

The reviewer thanks the authors for the time and attention paid to the comments from all reviewers. Following the below mandatory changes, the paper will be ready to accept for publication.

1. The use of hybrid and heterogenous terms are well-known. All that is being asked is that the manuscript very clearly uses the generally accepted definitions with clarity so that the reader is clear about the results. I believe the authors have replied with the correct definitions. In the abstract and all places in the manuscript, make the clear distinctions:
 - a. Only claim heterogeneous when referring to the lithium niobate bonded to the silicon nitride.
 - b. Claim hybrid when talking about the laser since the gain chip is butt-coupled.
2. It is hard to tell from the following response if the authors agree to the above or not. It is a confusing statement (a). There are too many acronyms, newly defined platforms ... what is meant by the red sentence, please be more specific for a general reader.
 - a. "The LNOD platform is a hybrid Si₃N₄/LiNbO₃ platform."
3. The updated comparison of tunable lasers table is nice and adds to the quality of the paper.
4. As long as the claims are limited to the truly novel claims I am good with the updates.
5. I find the argument about loss and Q to limiting. Papers that are accepted and published in a peer-reviewed journal should not be debated unless there is a response or errata to a published work or unless able to prove the published numbers wrong. I do not agree that the Loncar paper can be discounted based on only one data point. I do appreciate and agree with the statistical analysis approach, but do not agree that the claims should be made about discounting a published number. In any case, it is not the main novelty or claim of this manuscript and I encourage the authors to let this one pass in the final revision.
6. The new table along these lines is a good addition.
7. I cannot agree with the following statement: "However, the direct comparison of these results to our work is not suitable for the following reasons. The authors need to compare with all related works since the claims are being made beyond just the fast tunable low linewidth. The comparison is suitable and needs to be included.
8. The following is not completely true. 4.6 million is not on-par or better than most pure Si₃N₄ integrated photonic circuits. Much lower losses (and higher Q) have been reported for the pure silicon nitride waveguide platform. Many orders of magnitude better Q and lower losses – 422 million in this same journal (Nature Comm) and 720 million (Optics Letters), which are also low confinement, much higher than even the examples mentioned below (e.g. the Jin work reported about 200 million). Any statement or claim per the below needs to be removed from the paper with respect to other purely silicon nitride circuits, or put into correct context with the appropriate citations. As such, appropriate credit for previous work in ultra high Q and ultra low losses needs to be given in the manuscript to other works which have record performance, as these are all

considered purely silicon nitride circuits (the Jin work is not).

a. As for the Q of the resonator, our value of 4.6×10^6 for a lithium niobate device is an excellent value, on par with or better than most pure Si₃N₄ integrated photonics circuits, a significantly lower loss than silicon photonics, and surpassed only by platforms with very low confinement [Jin et al. Nat. Phot. 2021], our own Damascene platform [Liu, J. et al., Nat. Commun. 12, 2236 (2021)], or wide over-moded Si₃N₄ waveguides [Ji, X., Jang, et al., Laser & Photonics Reviews 2020, 15, 2000353].

Referee #2 (Remarks to the Author):

A. Summary of the key results

The work submitted for review presents the performance of a hybrid integrated laser source realized combining a Distributed Feedback Grating (DFB) laser in InP with a ring resonator defined in the Lithium Niobate on Damascene Si₃N₄ (LNOD) platform. The external ring resonator cavity provides a weak back-reflected wave to the DFB, causing injection locking regime in specific conditions. The laser is tuned in locking regime by changing the resonance frequency of the ring resonator by the electro-optic Pockel effect. The authors claim to reach an ultra-fast tuning speed of 12 PHz/s over a range of 600 MHz, while keeping an intrinsic laser linewidth of 3 kHz. The laser swept source is then deployed on a coherent LIDAR experiment using a Frequency-Modulated Continuous Wave (FMCW) scheme, demonstrating a resolution of 15 cm in a 3 m range.

B. Originality and significance: if not novel, please include reference

The main novelty, according to the authors, lies in the realization of integrated laser using their LNOD platform. Their platform allows to combine high-confinement silicon-nitride waveguides with the electro-optic material lithium niobate, so to obtain compact, high-quality factor and ultrafast tuning. The concept is of extreme interest and should be pushed further.

C. Data & methodology: validity of approach, quality of data, quality of presentation

The work and data are well described. Data plots are clear and many data are available.

D. Appropriate use of statistics and treatment of uncertainties

The experimental data are represented together with their deviation from the intended value. Such data give clear indication of the performance of the device and the range of applicability.

E. Conclusions: robustness, validity, reliability

The conclusions are valid, and consistent with what was exposed throughout the paper. They discuss possible application and future paths for this research.

F. Suggested improvements: experiments, data for possible revision

At the current stage, I don't see the need for other experiments to carry on besides improving the performance of the device. Would it be possible to have some data on the back-reflection coefficient with varying Q factor or coupling factor?

G. References: appropriate credit to previous work?

The work provides appropriate credit to previous work satisfactorily.

H. Clarity and context: lucidity of abstract/summary, appropriateness of abstract, introduction and conclusions

The abstract claim is clear, highlights the points of strength with a logical structure. The introduction gets very technical from the beginning and skips some details that may not be obvious during the

exposition. The conclusion is clear and cut.

Referee #3 (Remarks to the Author):

The authors addressed all technical reviewer remarks. In particular the supplementary material became much more profound with regard to addressing the state of the art.

The technical quality of the work is without any doubt very high. The only question is whether relevance and impact justify a publication in Nature. Here the assessment is finally with the Editors in view of all comments received.

Author Rebuttals to First Revision:

REVIEWER COMMENTS

We are grateful that three Referees have again seen our manuscript. We appreciate reviews by the reviewers and have modified the manuscript to highlight changes we made to address concerns of Reviewer #2 and Reviewer #3. Here, we present a point-by-point reply (in **blue**) to the reviewers' comments (in **black**), as well as the action taken (in **red**).

Referees' comments:

Referee #1 (Remarks to the Author):

Review of Nature 2021-10-16184A-Z

The reviewer thanks the authors for the time and attention paid to the comments from all reviewers. Following the below mandatory changes, the paper will be ready to accept for publication.

1. The use of hybrid and heterogenous terms are well-known. All that is being asked is that the manuscript very clearly uses the generally accepted definitions with clarity so that the reader is clear about the results. I believe the authors have replied with the correct definitions. In the abstract and all places in the manuscript, make the clear distinctions:

- a. Only claim heterogeneous when referring to the lithium niobate bonded to the silicon nitride.
- b. Claim hybrid when talking about the laser since the gain chip is butt-coupled.

We double-checked the manuscript's main text and confirmed that we use the term "heterogeneous" only for LNOD integration and "hybrid" only for the laser.

2. It is hard to tell from the following response if the authors agree to the above or not. It is a confusing statement (a). There are too many acronyms, newly defined platforms ... what is meant by the red sentence, please be more specific for a general reader.

- a. "The LNOD platform is a hybrid Si₃N₄/LiNbO₃ platform."

We used this sentence only in the review reply and not in the main manuscript. In the main manuscript, we use the generally accepted definition – of heterogeneous integration for bonding LiNbO_3 to Si_3N_4 . We double-checked the main text and found no issues in this respect.

3. The updated comparison of tunable lasers table is nice and adds to the quality of the paper.

We appreciate that the Reviewer gave a positive evaluation of the comparison table and revised text.

4. As long as the claims are limited to the truly novel claims I am good with the updates.

5. I find the argument about loss and Q to limiting. Papers that are accepted and published in a peer-reviewed journal should not be debated unless there is a response or errata to a published work or unless able to prove the published numbers wrong. I do not agree that the Loncar paper can be discounted based on only one data point. I do appreciate and agree with the statistical analysis approach, but do not agree that the claims should be made about discounting a published number. In any case, it is not the main novelty or claim of this manuscript and I encourage the authors to let this one pass in the final revision.

We agree with the Reviewer on this point. In no case do we discount the work of the Loncar group - the paper is cited in the main text and also present in the comparison table 1.

6. The new table along these lines is a good addition.

We thank the Referee for this acknowledgment.

7. I cannot agree with the following statement: “However, the direct comparison of these results to our work is not suitable for the following reasons. The authors need to compare with all related works since the claims are being made beyond just the fast tunable low linewidth. The comparison is suitable and needs to be included.

We agree with the Referee that the comparison to W. Jin, et al. “Hertz-linewidth semiconductor lasers using CMOS-ready ultra-high-Q microresonators,” (Nature Photonics, 2021) is relevant for our laser. We include it into the Extended Data table 1 and cite in the main text.

Action taken: We added comparison to Jin et al. to the Extended Data table 1.

8. The following is not completely true. 4.6 million is not on-par or better than most pure Si_3N_4 integrated photonic circuits. Much lower losses (and higher Q) have been reported for the pure silicon nitride waveguide platform. Many orders of magnitude better Q and lower losses – 422 million in this same journal (Nature Comm) and 720 million (Optics Letters), which are also low confinement, much higher than even the examples mentioned below (e.g. the Jin work reported about 200 million). Any statement or claim per the below needs to be removed from the paper with respect to other purely silicon nitride circuits, or put into correct context with the appropriate citations. As such, appropriate credit for previous work in ultra high Q and ultra low losses needs to be given in the manuscript to other works which have record performance, as these are all considered purely silicon nitride circuits (the Jin work is not).

a. As for the Q of the resonator, our value of 4.6×10^6 for a lithium niobate device is an excellent value, on par with or better than most pure Si₃N₄ integrated photonics circuits, a significantly lower loss than silicon photonics, and surpassed only by platforms with very low confinement [Jin et al. Nat. Phot. 2021], our own Damascene platform [Liu, J. et al., Nat. Commun. 12, 2236 (2021)], or wide over-moded Si₃N₄ waveguides [Ji, X., Jang, et al., Laser & Photonics Reviews 2020, 15, 2000353].

We agree with the Reviewer that the comparison statements should be put into the correct context. We added citations to high-Q low confinement Si₃N₄ 422 million (Nature Comm) and 720 million (Optics Letters).

Action taken: We added citations to high-Q low confinement Si₃N₄ work to the main manuscript [references 26,27,28,29].

Referee #2 (Remarks to the Author):

A. Summary of the key results

The work submitted for review presents the performance of a hybrid integrated laser source realized combining a Distributed Feedback Grating (DFB) laser in InP with a ring resonator defined in the Lithium Niobate on Damascene Si₃N₄ (LNOD) platform. The external ring resonator cavity provides a weak back-reflected wave to the DFB, causing injection locking regime in specific conditions. The laser is tuned in locking regime by changing the resonance frequency of the ring resonator by the electro-optic Pockel effect. The authors claim to reach an ultra-fast tuning speed of 12 PHz/s over a range of 600 MHz, while keeping an intrinsic laser linewidth of 3 kHz. The laser swept source is then deployed on a coherent LIDAR experiment using a Frequency-Modulated Continuous Wave (FMCW) scheme, demonstrating a resolution of 15 cm in a 3 m range.

B. Originality and significance: if not novel, please include reference

The main novelty, according to the authors, lies in the realization of integrated laser using their LNOD platform. Their platform allows to combine high-confinement silicon-nitride waveguides with the electro-optic material lithium niobate, so to obtain compact, high-quality factor and ultrafast tuning. The concept is of extreme interest and should be pushed further.

C. Data & methodology: validity of approach, quality of data, quality of presentation

The work and data are well described. Data plots are clear and many data are available.

D. Appropriate use of statistics and treatment of uncertainties

The experimental data are represented together with their deviation from the intended value. Such data give clear indication of the performance of the device and the range of applicability.

E. Conclusions: robustness, validity, reliability

The conclusions are valid, and consistent with what was exposed throughout the paper. They discuss possible application and future paths for this research.

We appreciate that the Reviewer gave a positive evaluation of the revised manuscript.

F. Suggested improvements: experiments, data for possible revision

At the current stage, I don't see the need for other experiments to carry on besides improving the performance of the device. Would it be possible to have some data on the back-reflection coefficient with varying Q factor or coupling factor?

We present back reflection characterization for 3 different chips from D67_01 wafer in Supplementary Figure 8. All resonances are undercoupled. Backreflection of only up to 4% can be observed (see also Extended Data Figure 2).

Action taken: We added back reflection data along with cavity transmission measurements in Extended Data figure 8 for 3 different samples.

G. References: appropriate credit to previous work?

The work provides appropriate credit to previous work satisfactorily.

H. Clarity and context: lucidity of abstract/summary, appropriateness of abstract, introduction and conclusions

The abstract claim is clear, highlights the points of strength with a logical structure. The introduction gets very technical from the beginning and skips some details that may not be obvious during the exposition. The conclusion is clear and cut.

We thank the Reviewer for the report and appreciate that the abstract and the conclusion are recognized.

Referee #3 (Remarks to the Author):

The authors addressed all technical reviewer remarks. In particular the supplementary material became much more profound with regard to addressing the state of the art.

The technical quality of the work is without any doubt very high. The only question is whether relevance and impact justify a publication in Nature. Here the assessment is finally with the Editors in view of all comments received.

We thank the Referee for the acknowledgment of the technical quality of our work and the revised supplementary material, and the main text.